# Structural dynamics of AAA + ATPase Drg1 and mechanism of benzo-diazaborine inhibition

Chengying Ma[1,2,4], Damu Wu[1,4], Qian Chen[1] & Ning Gao ®[1,2,3] ✉

The type II AAA + ATPase Drg1 is a ribosome assembly factor, functioning to release Rlp24 from the pre-60S particle just exported from nucleus, and its activity in can be inhibited by a drug molecule diazaborine. However, molecular mechanisms of Drg1-mediated Rlp24 removal and diazaborine-mediated inhibition are not fully understood. Here, we report Drg1 structures in different nucleotide-binding and benzo-diazaborine treated states. Drg1 hexamers transits between two extreme conformations (planar or helical arrangement of protomers). By forming covalent adducts with ATP molecules in both ATPase domain, benzo-diazaborine locks Drg1 hexamers in a symmetric and non-productive conformation to inhibits both inter-protomer and inter-ring communication of Drg1 hexamers. We also obtained a substrate-engaged mutant Drg1 structure, in which conserved pore-loops form a spiral staircase to interact with the polypeptide through a sequence-independent manner. Structure-based mutagenesis data highlight the functional importance of the pore-loop, the D1-D2 linker and the inter-subunit signaling motif of Drg1, which share similar regulatory mechanisms with p97. Our results suggest that Drg1 may function as an unfoldase that threads a substrate protein within the pre-60S particle.

AAA + ATPases (ATPases associated with a variety of cellular activities) are a superfamily of proteins with various functions in remodeling their substrates in different subcellular localization, ranging from protein quality control, membrane trafficking to DNA/RNA unwinding[1–6]. Many AAA + ATPases are dedicated to the unfolding of proteins and the disassembly of macromolecular complexes. These protein-remodeling AAA + ATPases generally consist of two types: Type I, such as Katanin and Spastin which contribute to microtubule dynamics[7], contains an N-terminal domain (NTD) and an ATPase domain, whereas Type II, such as N-ethylmaleimide sensitive factor (NSF), p97/Cdc48, Hsp104 and ClpB[8–14], contains an NTD and two ATPase domains (D1 and D2). The NTD often binds to adapter proteins, which are responsible for substrate recognition[5,15]. The adapters can localize, regulate, and determine the specific functions of

AAA + ATPases[16–18]. Over recent years, based on extensive structural and functional studies on several type II AAA + proteins, the mechanistic understanding of AAA + proteins in protein unfolding has been greatly advanced[2,8,9,11,12,14,19–22]. Especially, the structural studies on AAA + ATPase hexamers with their native substrates revealed the molecular basis of substrate recognition and processing with a hand-over-hand mechanism[2,5,23].

Eukaryotic ribosome biogenesis is a complicated and energy-consuming process that requires more than 200 ribosome assembly factors[24–27]. Three AAA + ATPases are involved in the assembly of the large ribosomal subunit, including Rix7, Rea1, and Drg1, which are located in nucleolus, nucleoplasm, and cytoplasm, respectively[28]. Rea1, as a member of dynein-related AAA + ATPase subfamily, is involves in two steps of the 60S maturation, including the release of Nop7 and

[1]State Key Laboratory of Membrane Biology, Peking-Tsinghua Joint Center for Life Sciences, School of Life Sciences, Peking University, 100871 Beijing, China. [2]Changping Laboratory, 102206 Beijing, China. [3]National Biomedical Imaging Center, Peking University, 100871 Beijing, China. [4]These authors contributed equally: Chengying Ma, Damu Wu. ✉e-mail: gaon@pku.edu.cn

Rsa4 in nucleolus and nucleoplasm, respectively[29,30]. Both Rix7 and Drg1 belong to type II AAA + proteins. Rix7 was proposed to act as the earliest AAA + ATPase in the maturation pathway to catalyze the removal of Nsa1 from the nucleolar pre-60S ribosomal particles[31]. Drg1 initiates the cytoplasmic pre-60S maturation, and is necessary for releasing several shuttling factors Nog1, Rlp24, Tif6, and Mrt4 and export factors Mex6/Mtr2 from the cytoplasmic pre-60S particles[32,33]. Specifically, the NTD of Drg1 was reported to interact with the C-terminal domain (CTD) of Rlp24, and this interaction in turn enhances the ATPase activity of Drg1[32]. The function of Drg1 may also require a nuclear pore protein Nup116, which was identified as a Drg1 interactor through a yeast two-hybrid analysis[32]. Nup116 might have a potential adapter function for the release of Rlp24 from pre-60S particles. Presumably, the removal of Rlp24 directly by Drg1 triggers the simultaneous or subsequent dissociation of other biogenesis factors that interact with Rlp24 on pre-60S particles[34].

Drg1 is an essential protein in *Saccharomyces*[35], and its mammalian homolog SPATA5 is associated with spermatogenesis by regulating mitochondrial functions[36,37]. SPATA5 mutations in human have been linked to a disease condition, Epilepsy, Hearing Loss, and Mental Retardation Syndrome[38]. Yeast *AFG2* encoding Drg1 was also identified as one of the genes related to the sensitivity of the cells towards diazaborine treatment[39]. Diazaborine is a heterocyclic boron-containing compound targeting fatty acid and phospholipid biosynthesis in Gram-negative bacteria[40,41]. Diazaborine inhibits the ATP hydrolysis of Drg1 and therefore leads to accumulation of shuttling proteins on the pre-60S particles in the cytoplasm[32,33,42,43]. Importantly, diazaborine resistance mutations were found to locate close to the D2 ATPase site of Drg1[33]. Recently, two cryo-EM studies have reported the structural characterization of yeast Rix7 and Drg1[22,44], revealing a conserved hexameric architecture for the two proteins ATPases, highly similar as those of Cdc48/p97[8,11,12,14]. The structure of Drg1 was solved in the presence of diazaborine which explained the structural basis of the drug action on the D2 domain[44]. However, neither of them provided the high-resolution structures for the NTDs of these two proteins, and the structure of Drg1[44] was not in the substrate-processing state.

In this work, we provide a comprehensive structural characterization of yeast Drg1 hexamers in the presence of ADP, AMPPNP, a mixture of ADP/AMPPNP/benzo-diazaborine or a mixture of ATP/benzo-diazaborine. We also obtain a structure of a mutant form of Drg1 (E346Q/E617Q) hexamer, featuring a polypeptide substrate in the central channel with staggered interactions with the pore loops of Drg1 ATPase domains. These structures provide insights into the general mechanisms of substrate unfolding and translocation for type II AAA + proteins, and the molecular basis for understanding the specific function of Drg1 in ribosome assembly. In addition, our results show that benzo-diazaborine binds to the ATPase sites of both D1 and D2 domains, and locks the hexamer in a more symmetrical conformation. As one of the highly potent inhibitors in the diazaborine family compounds against bacteria[45], our data thus provide critical information for improvement of benzo-diazaborine chemicals as anti-fungal drugs[46].

## Results

### Structural determination of Drg1 hexamers in different nucleotide binding states

To understand the structural dynamics of *S. cerevisiae* Drg1 hexamer, we assembled Drg1 hexamers in vitro in the presence of excessive ADP or AMPPNP (a non-hydrolysable analog of ATP). To obtain possible hybrid nucleotide bound states that best mimic the ATPase hydrolysis cycle of Drg1 and to illustrate the inhibition mechanism of benzo-diazaborine, we also incubated Drg1 with a mixture of ADP, AMPPNP, and benzo-diazaborine, as well as with a mixture of ATP and benzo-diazaborine. These samples were subjected to electron microscopic analysis. Negative staining electron microscopy confirmed the

hexameric forms of all these Drg1 samples (Supplementary Fig. 1). The structures of Drg1 hexamers (Fig. 1) from these different samples were subsequently determined using cryo-EM at resolutions of 3.5 to 5.6 Å (Supplementary Figs. 2–5). In general, Drg1 hexamers are high dynamic and exhibit different modes of motion. All structures display a typical three-layered architecture, similar to those of type II AAA + hexamers, such as p97/VCP and Cdc48[8,19], and the NTD, D1, and D2 rings show an apparent deviation from a perfect sixfold symmetry. Multiple rounds of 3D classifications suggest that the conformational changes of Drg1 hexamers are likely continuous, as different 3D classes tend to have comparable numbers of particles (Supplementary Figs. 2–5). Comparisons of these structures indicate that Drg1 hexamers transit between two extreme conformations, a planar and a helical state. While the six subunits in the planar or planar-like states are arranged more symmetrically, the subunits in the helical states adopt a spiral configuration to varying extents. In summary, these cryo-EM analyses provide an ensemble of structures that reflect intrinsic properties of Drg1 hexamers.

Next, we prepared a mutant form of Drg1 with two mutations (E346Q and E617Q) located in Walker B motif of the D1 and D2 domains, respectively[1]. These two mutations dramatically slow down ATP hydrolysis but not ATP binding to the active centers of the D1 and D2 domains[32]. Mutant Drg1 hexamers were incubated with excessive ATP for stable hexamer formation before cryo-EM sample preparation. The cryo-EM structure of mutant Drg1 hexamers was determined at a much-improved resolution of 3.5 Å (Fig. 2). Similar as in the structural study of Rix7[22], the introduction of the Walker B double mutation allowed the capture of an endogenous peptide within the central channel of Drg1 hexamer during purification. Upon the placement of the polypeptide in an upright position, the six protomers clearly form spiral staircases around the peptide (Fig. 2d). This substrate binding should have stabilized the hexamer and likely reduced its conformational space. This is probably the reason why the structure of the mutant hexamers could be resolved at a higher resolution, allowing the atomic modeling of most of the Drg1 sequence.

Based on the high-resolution map of the mutant hexamer, we built an atomic model of Drg1 from residues 29 to 777 (Fig. 3). The residues 29–288, 250–499, and 521–77 constitute the NTD, D1, and D2, respectively. A surprising finding is that the NTD structure of Drg1 is highly similar to those of p97 and Cdc48 (Fig. 4b–f), despite of their low sequence homology (Fig. 4a). Compared to Cdc48, the orientation of the NTD relative to the D1 domain in the Drg1 hexamer is different, displaying a large rotation (Fig. 4g–i).

The D1 and D2 domains can be divided into two subdomains (an α/β core and an α-helical lid subdomain). The α/β core subdomain is composed of a five-stranded β-sheet and five α-helices in the order of α0-β1-α1-β2-α2-β3-α3-β4-α4-β5. The α-helical lid is composed of four α-helices (α5-α6-α7-α8) (Fig. 3a). The functional motifs of AAA + domains, such as Walker A, Walker B, Sensor I, Sensor II, arginine finger (AF), and pore loops I and II (PL-I, PL-II), are all well resolved (Fig. 3b–d). The models of Drg1 hexamers treated with ADP, AMPPNP, ADP/AMPPNP/benzo-diazaborine, and ATP/benzo-diazaborine were similarly obtained, by rigid-body fitting of multiple copies of the Drg1 model (with manual adjustment and rebuilding when necessary).

### Structures of Drg1 hexamer in the absence of substrate

A distinct feature for Drg1 hexamers from other AAA + hexamers, such as p97/Cdc48 and NSF, is that the NTDs do not show a large-scale, nucleotide-dependent axial displacement (Figs. 1 and 2). For Cdc48 and p97, their NTDs form a plane with the D1 domains in the presence of ADP (co-planar with the D1 ring) and turn into an "up" conformation upon binding to ATPγS[8,11,12,19]. In all our structures, the NTD rings of Drg1 hexamers are always in "up" conformations (Figs. 1 and 2 and Supplementary Figs. 2–5).

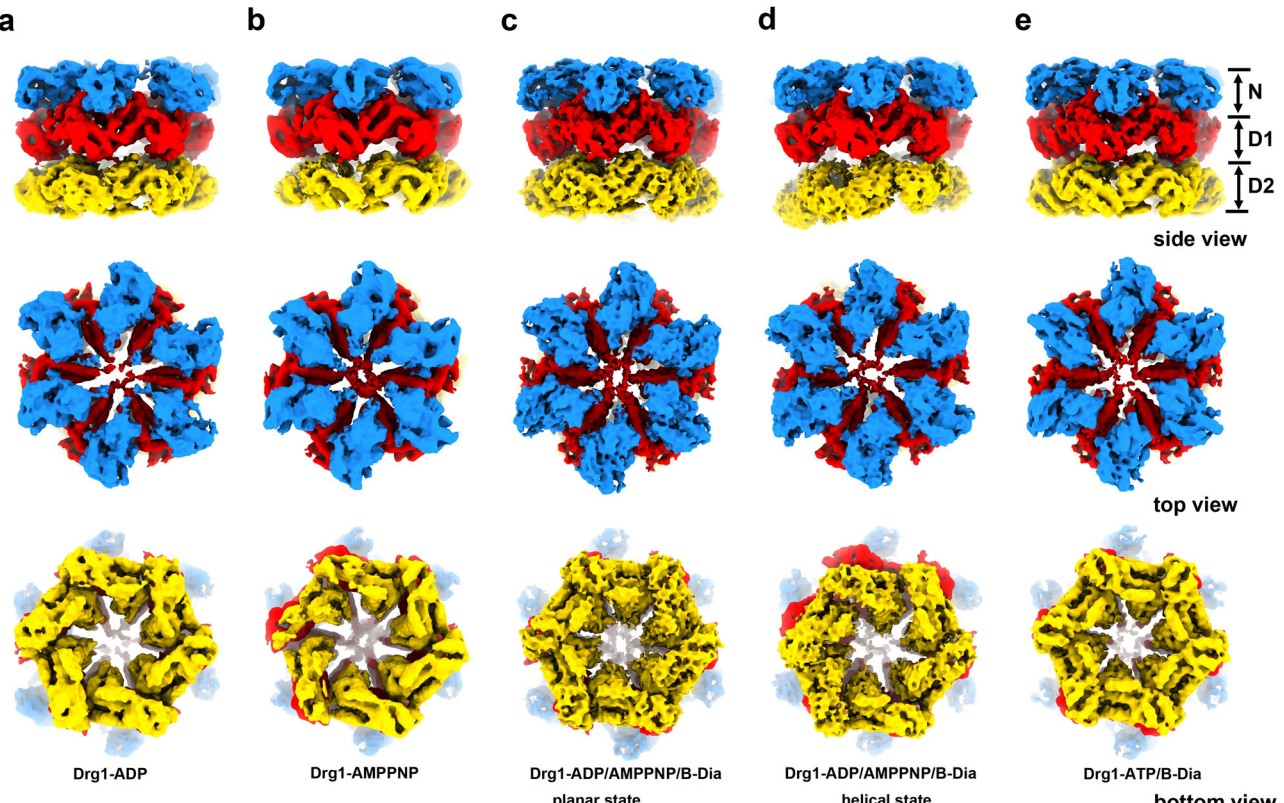

**Fig. 1 | Representative cryo-EM structures of wild type Drg1 hexamers in different nucleotide and benzo-diazaborine binding states. a–e** Representative cryo-EM maps of Drg1 hexamers treated with ADP (**a**), AMPPNP (**b**), ADP/AMPPNP/ benzo-diazaborine (**c**, **d**) or ATP/benzo-diazaborine (**e**). The density maps are shown in the side, top (NTD) and bottom (D2) views. The NTD, D1, and D2 are colored dodger blue, red, and orange, respectively. B-Dia, benzo-diazaborine.

To better understand the structural dynamics of Drg1 hexamers related to the nucleotide hydrolytic cycle, we analyzed the structures of Drg1 hexamers obtained from three different samples (ADP, AMPPNP, ADP/AMPPNP/benzo-diazaborine). In these datasets, only a small fraction of particles displays an approximate six-fold symmetry (Supplementary Fig. 2–5). Most 3D classes adopt asymmetrical conformations, with one or two subunits more flexible than others. In addition, a few 3D classes also show highly flexible D2 rings (Supplementary Figs. 2a, 3a, 4a). In contrast, for the ATP/benzo-diazaborine treated hexamers, most of the 3D classes are more symmetrical (Fig. 1e and Supplementary Fig. 5c).

Taking the dataset of ADP/AMPPNP/benzo-diazaborine as an example, we performed deep 3D classification and analyzed the conformational differences between its planar and helical states (Fig. 1c, d and Supplementary Figs. 4 and 6). While the NTD and D1 rings are roughly superimposable, the D2 rings display a very large rotation between two states (Supplementary Figs. 4a and 6). These results indicate that the conformational cycling of Drg1 hexamers would generate a large-scale inter-ring rotation, in addition to the axial displacement of individual subunits.

### Distinct nucleotide occupation of the D1 and D2 rings in Drg1 hexamers

An intriguing observation is that the nucleotide binding pockets of D1 domains in the structures of different datasets are all occupied, either by ATP or AMPPNP (Supplementary Figs. 7a–10a), even for the Drg1-ADP sample (Supplementary Fig. 7a). Although the structure from the Drg1-ADP dataset was not resolved at atomic resolution, but the local densities suggest the presence of ATP in these D1 sites. Therefore, this co-purification of ATP with Drg1 implies that the D1 ATPase site likely has a high affinity for ATP and a low hydrolysis activity. A similar observation was also made on p97: its D1 domain has a higher

nucleotide affinity than the D2 domain[47,48]. Although the nucleotide affinity and the ATPase activity of the D1 and D2 domains of Drg1 are not explicitly examined, some data did reveal sharply different enzymatic properties for the two domains[32,33,49].

There is no obvious nucleotide occupation in the D2 domains of the structures from the ADP and AMPPNP treated samples (Supplementary Figs. 7b and 8b). The nucleotide binding to the D2 domains was only seen in structures from benzo-diazaborine treated samples, including the ADP/AMPPNP/benzo-diazaborine and ATP/benzo-diazaborine datasets (Supplementary Figs. 9b, 10b, and S13). While the six D2 ATPase sites in the structure of the ATP/benzo-diazaborine dataset are fully occupied by ATP, the six D2 sites in the structures from the ADP/AMPPNP/benzo-diazaborine dataset (Supplementary Figs. 9b and 10b) are not saturated and appear to be partially occupied by ADP. These observations again suggest different affinity and complex regulation on the ATP/ADP binding to the D1 and D2 domains. Nevertheless, these results are in accordance with previous functional data that Drg1 D1 mutant (E346Q) does not affect growth, and has little effect on the release of shuttling proteins from pre-60S particles, whereas Drg1 D2 mutation (E617Q) is lethal[32]. Therefore, like many type-II AAA hexamers, the primary role of one high-affinity ATP binding site (D1 in p97, D2 in NSF, and D1 in Drg1) is likely to facilitate the nucleotide-dependent hexamer formation (reviewed in ref. 5). In summary, our data suggest a complex regulation on the nucleotide binding to the 12 ATPase sites in a single hexamer.

### Structure of Drg1 hexamer captured in processing peptide substrate

In the structure of the mutant Drg1 hexamer (Drg1$^{E346Q/E617Q}$-ATP), a polypeptide was found in the central channel, which passes through the central pores of Drg1 hexamers formed by the PLs of the D1 and D2 rings (Figs. 2 and 5). This peptide should be a non-specific substrate of

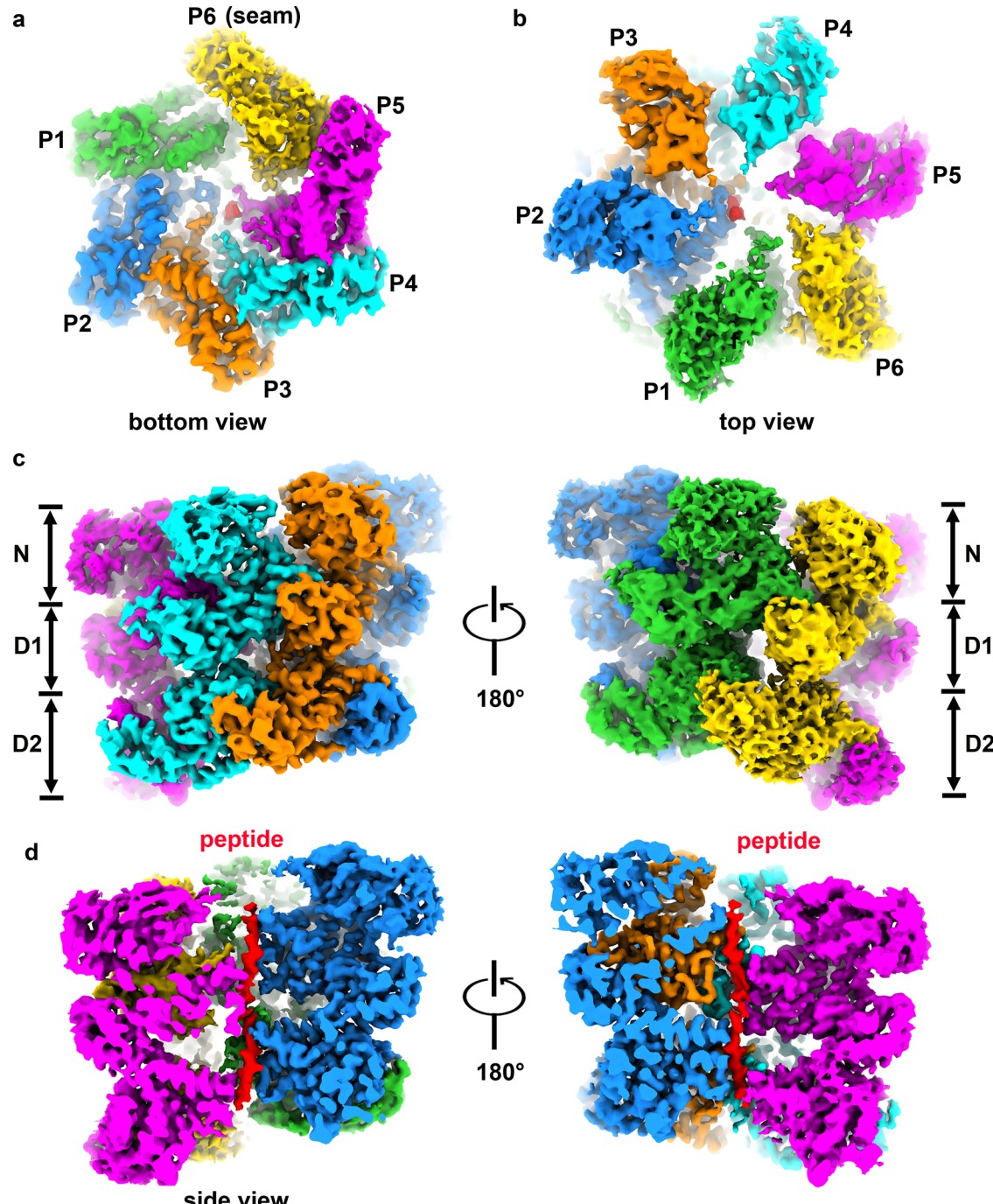

**Fig. 2 | Cryo-EM structure of the double mutant Drg1$^{E346Q/E617Q}$ hexamer in the presence of ATP. a, b** Cryo-EM map of the double mutant Drg1$^{E346Q/E617Q}$ hexamer in the presence of ATP, viewed from the D2 (**a**) and NTD (**b**) directions. **c** Side view of the double mutant Drg1$^{E346Q/E617Q}$ hexamer, highlighting its helical configuration of protomers. **d** Cross-section view shows the trapped substrate within the central channel of the double mutant Drg1$^{E346Q/E617Q}$ hexamer.

*E. coli* origin that was processed by Drg1 hexamers in *E. coli* cells and therefore co-purified with mutant Drg1 hexamers. The density of the main chain of this peptide was well resolved such that we could build a poly-alanine model (23 residues) for it. The peptide spans about 65 Å length of the pore (Fig. 5d), and is similar to the one in the Rix7-structure which also contains a threaded peptide in the central channel[22].

The six protomers of Drg1 form a spiral staircase to accommodate the peptide. Among the six protomers of Drg1 (named P1-P6), five are apparently more stable and display a higher resolution in the density map (Fig. 2 and Supplementary Fig. 5f–j). Each protomer was then subjected to a mask-based 3D refinement. The improved local maps of

P1 to P6 confirm that P6 is relatively unstable compared to the other five protomers. Thus, P6 of the mutant Drg1 hexamer should be the "seam" subunit reported in the structures of Cdc48/p97 and Rix7 in substrate-engaged states (Supplementary Fig. 11)[2,11,12,14,22]. Although the density of P6 is not well resolved at atomic resolution, it is clear enough to conclude that P6 is relatively detached from the peptide and other protomers in the hexamer (Figs. 2 and 5b, c).

When the trapped peptide is used as a vertical ruler, the six spirally arranged protomers display different axial displacements (Supplementary Fig. 12a), with P5 being the lowest. When the six protomers were superimposed with the D1 domains as reference, a few principles regarding the inter-domain conformational changes of Drg1 subunits

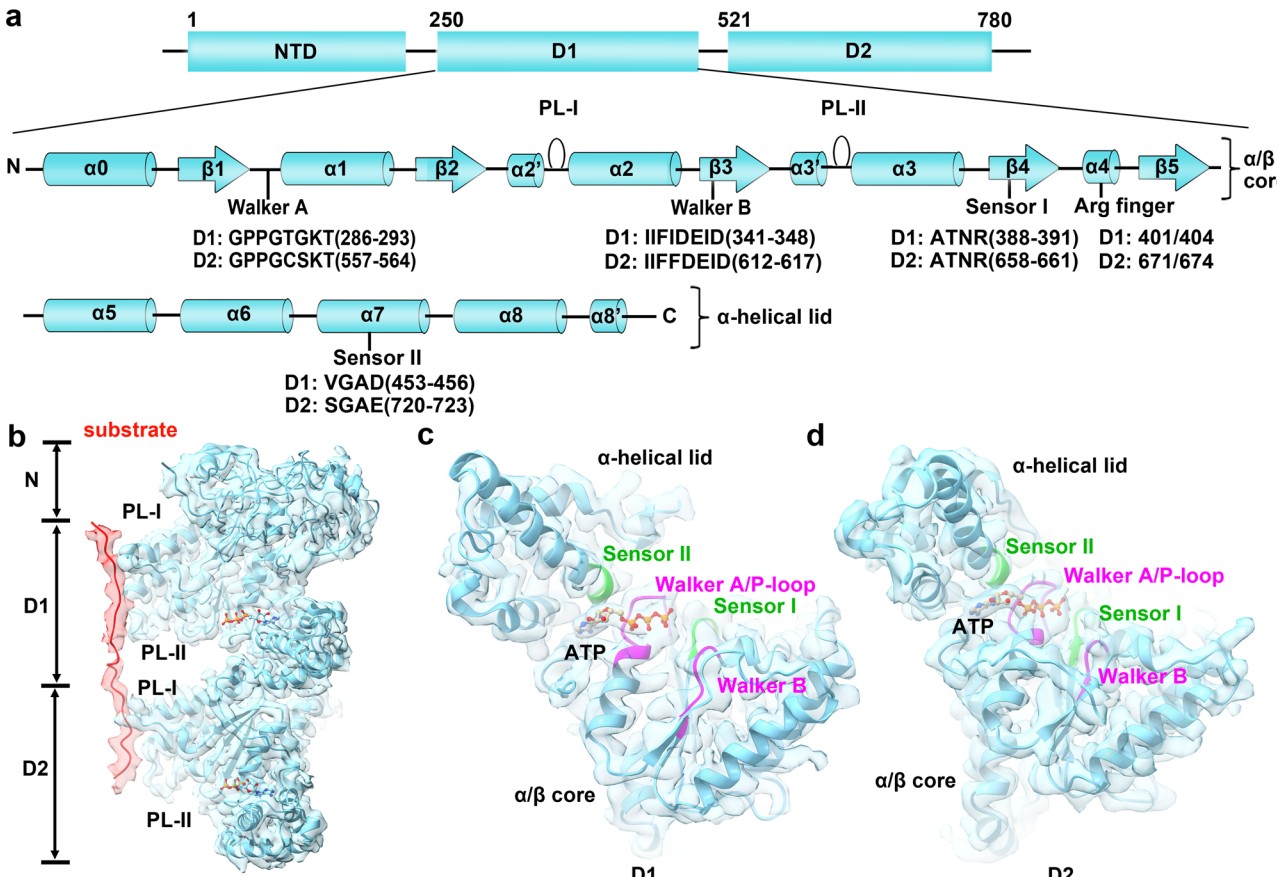

**Fig. 3 | Atomic model and domain organization of Drg1. a** Domain organization and secondary structure of *S. cerevisiae* Drg1. The AAA + domain contains an α/β core subdomain (α0-β5) and an α-helical lid subdomain (α5-α8). The highly conserved motifs including Walker A, Walker B, sensor I, sensor II and arginine fingers, pore loops (PL-I and PL-II) are shown as indicated. **b** The atomic model of Drg1 is shown in cartoon representation, superimposed with the density map of one Drg1 protomer (P3). The substrate is colored red and the PLs are labeled. ATP molecules in the ATPase center of the D1 and D2 domains are highlighted in stick models. **c, d** The atomic models of the D1 and D2 domains, superimposed with the cryo-EM map of the protomer P3. The functional motifs including Walker A, Walker B, sensor I, and sensor II are colored separately. ATP is highlighted in stick model.

could be derived. First, the connection between the NTD and D1 domains is relatively rigid, as the NTD-D1 displacements are always very small in these pair-wise comparisons (Supplementary Fig. 12b). Second, compared with the D2 domain of P1, the D2 domains of P2, P3, P4, and P5 display a similar rotational movement; but the D2 domain of P6 is in a distinct position, sharply different from other protomers (Supplementary Fig. 12b). Third, when the D2 domains were used as references of alignment, the same conclusion could be drawn that P6 is in an outlier conformation from the rest five (Supplementary Fig. 12c). Therefore, Drg1 protomers could take at least three major conformations, represented by P1, P2–P5, and P6, during the substrate processing cycle.

The D1 domains of all six protomers are in the ATP-bound state (Supplementary Fig. 13a–f), which is similar to the substrate engaged structures of Cdc48 and Rix7[11,12,22]. The AFs were proposed to coordinate ATP binding, nucleotide hydrolysis and conformational changes between adjacent protomers[50]. The AFs of the D1 domains (R401 and R404) were clearly resolved in P1-P5 (Supplementary Fig. 13a–f). The D1 AFs of P2, P3, P4, and P5 are in close to the bound ATP on the adjacent protomer. R404 residues in P3, P4 and P5 show a strong interaction with the γ-phosphate of ATP. In contrast, in P1 and P2, the γ-phosphate interaction is mainly mediated by R401, instead of R404. Also, as a seam subunit, the D1 AFs of P6 show no interaction with the adjacent ATP on P5.

In the D2 ring, protomers of P1 to P4 are in the ATP-bound state (Supplementary Fig. 13g–l). The D2 domain of P6 that is away from

the polypeptide is in the nucleotide-free state (Supplementary Fig. 13l). The D2 ATPase center of P5 is also occupied, although not fully resolved, and should be ATP as well (Supplementary Fig. 13k). The D2 AFs (R671 and R674) of P2, P3, P4, and P5 are close to the adjacent ATP (Supplementary Fig. 13g–j). Because P6 is relatively detached from the rest protomers, the D2 AFs of P6 show relatively larger distance from the bound ATP in the D2 domain of P5 (Supplementary Fig. 13k). Altogether, these results demonstrate that the nucleotide binding states of the two ATPase domains determine the conformations of Drg1 protomers, and that the subtle conformational changes at the protomer interface during ATP hydrolysis cycle could be translated into large-scale inter-protomer and inter-ring motions to drive the progressive movement of the substrate within the central channel.

### Conserved pore-loops are required for the function of Drg1 in vivo

The D1 PL-I of Drg1 (318-KYLG-321) contains a Tyr, whereas the D1 PL-Is of Cdc48/p97 (KMAG/KLAG, respectively) lack an aromatic residue. The D2 PL-I of Drg1 (589-KYVG-592) is highly similar to its D1 PL-I. In Cdc48/p97, the D2 PL-I is MWYG/MWFG, and the equivalent position is Trp instead of Tyr. It was previously shown that the introduction of an aromatic residue to the D1 PL-I of Cdc48 (M288Y, M288F, or M288W) conferred a lethal phenotype, whereas the Trp residue in the D2 PL-I of Cdc48 was strictly required for cell growth (W561A mutation was lethal)[51]

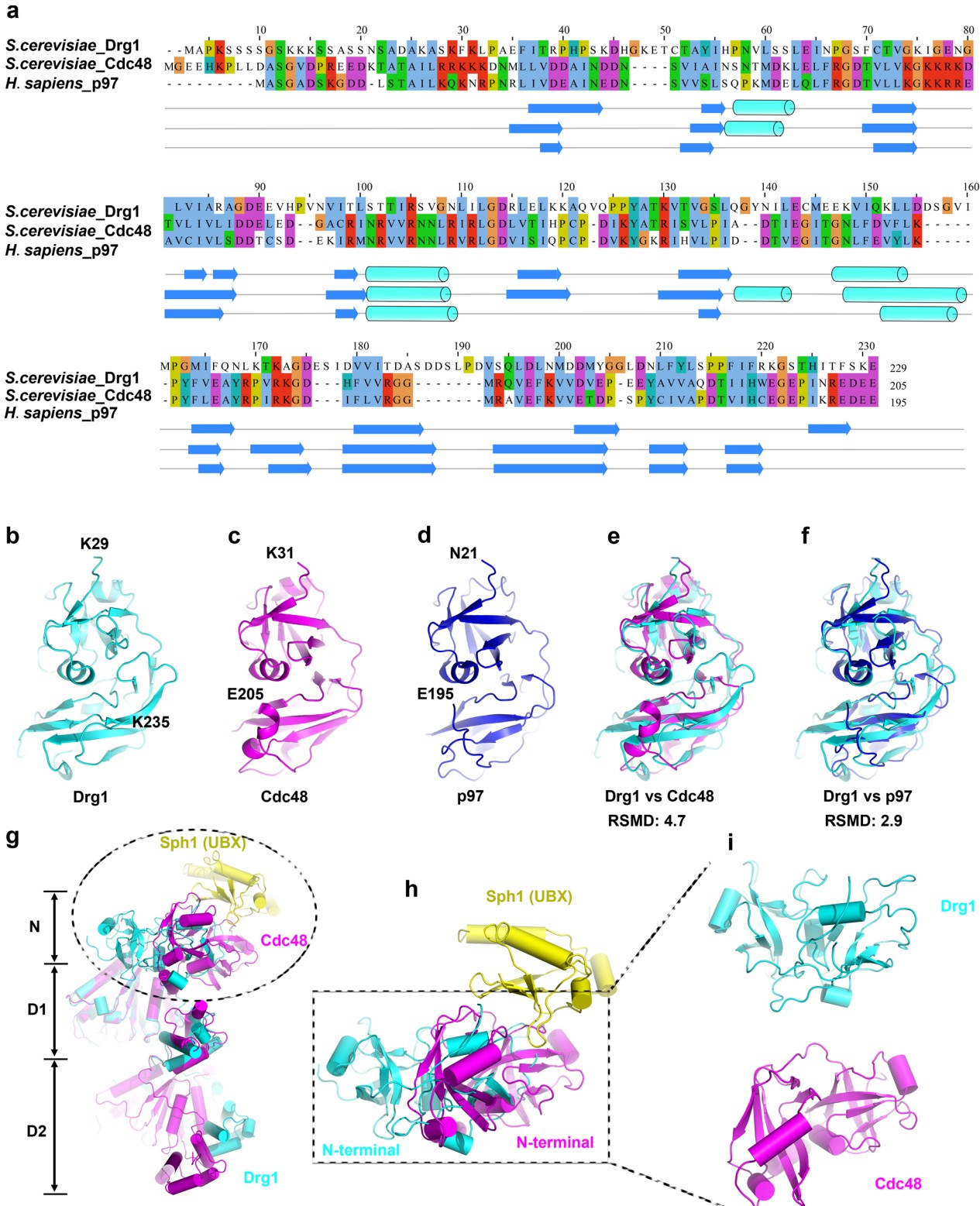

**Fig. 4 | Comparison of the NTDs from *S. cerevisiae* Drg1, Cdc48, and *H. sapiens* p97. a** Sequence alignment of the NTDs from *S. cerevisiae* Drg1, Cdc48, and *H. sapiens* p97. Secondary structures are also shown. **b–d** The atomic models of the NTD of *S. cerevisiae* Drg1, Cdc48 (PDB 6OPC)[11], and *H. sapiens* p97 (PDB 5FTK)[8]. **e** Superimposition of the NTDs of *S. cerevisiae* Drg1 and Cdc48 (RMSD 4.7 Å). **f** Superimposition of the NTD of *S. cerevisiae* Drg1 and *H. sapiens* p97 (RMSD 2.9 Å). **g** Superimposition of *S. cerevisiae* Drg1 and Cdc48 (PDB 6OPC). The two models are

aligned using the D1 ATPase domain as reference. The comparison shows that the NTDs of Drg1 and Cdc48 are in completely different orientations in the hexamer. The UBX domain of Sph1 (an adapter protein of Cdc48)[11] is shown and colored as indicated. **h**, **i** The enlarged view of (**g**) showing the orientation difference of the NTDs of Drg1 and Cdc48 relative to their D1 domains. Drg1, Cdc48, and UBX are colored as indicated.

In our Drg1 (double-mutation) structures, the PL-Is of the D1 and D2 interact with the polypeptide in the same fashion (Fig. 5), suggesting that they both contribute to the substrate binding and/or threading. The PLs (PL-I and PL-II) form upward spiral steps around the peptide and display an anticlockwise rotation pattern viewed from the D2 ring. The overall vertical displacements of the PLs in the D1 and D2 rings are about 22 Å and 28 Å, respectively (Fig. 5e), indicating that the PLs of two ATPase rings process the substrate with functional synergy. Specifically, while the polypeptide is held by PL-Is from four protomers (P2–P5) at the D1 layer, it is clamped by PL-Is from five protomers (P1–P5) at the D2 layer (Fig. 5b, c, e, f). The same observation holds true for the PL-IIs of the two rings. In general, while P2, P3, P4, and P5 show consistent interactions with the polypeptide at both the D1 and D2 layers, P1 only directly contact the polypeptide at the D2 ring. Therefore, in line with previous functional models of the PLs[2,23,52,53], our data clearly show that the PLs of Drg1 hexamer grip and move the substrate in a collaborative manner to facilitate substrate processing and translocation.

To verify the significance of PLs of D1 and D2, we introduced mutations to the PL-I and PL-II and evaluated their functional relevance using spotting assay and polysome profile analysis. Since a functional unit of Drg1 is a hexamer, overexpression of defective Drg1 mutants should have dominant negative effects. Indeed, as a control experiment, overexpression of the double mutant (E346Q/E617Q) dramatically impaired the cell growth and led to a polysome defect (halfmer). The appearance of halfmers on the right shoulder of polysome peaks is a typical phenotype of ribosome assembly defect. This is also accompanied by sharply elevated 40S/60S ratio on the polysome profile, which is another sign of the 60S assembly defect.

For the D1 and D2 PL-Is, two point-mutations were designed, Y319A and Y590A. For the PL-IIs, since they do not show loop residue-specific contact with the polypeptide (Figs. 5 and 6), we used three deletion mutations (based on distances measured in the structure), Δ362-365 or Δ357-361 for the D1 PL-II, and Δ628-632 for the D2 PL-II. Y319A, Δ362-365, and Δ357-361 are all deleterious to cell growth, but their effects are not as strong as those of the D2 mutants (Y590A, Δ628-632). Polysome profile analysis confirmed that all five mutants display a generally similar "halfmer" phenotype and a high 40S/60S ratio, indicating that the PL-Is and PL-IIs from the D1 and D2 domains all contribute to the in vivo function of Drg1.

Therefore, these results suggest that the aromatic residues in the PL-Is of the D1 and D2 are both functionally important, and confirmed the previous data[32] that the contribution from the D2 domain is larger than that of the D1 domain.

## The inter-subunit signaling motif contributes to the in vivo function of Drg1

The inter-subunit signaling (ISS) motif, a conserved region within the AAA + ATPase domain, plays an important role in regulating ATP hydrolysis[14,54,55]. Sequence alignment shows that a highly conserved ISS also exists in the D1 and D2 domains of Drg1 (D1:374-380, D2:644-650; Fig. 5g–j and Supplementary Fig. 11c).

Our structural analysis shows that the ISS motifs of the D1 and D2 domains in Drg1 are involved in the inter-subunit communication (Fig. 5h) and undergo highly similar conformational changes as those of p97[14]. In relatively tight inter-subunit interfaces, such as the P3-P4 interfaces in the D1 and D2 rings, the motif is in a stretched loop conformation (Fig. 5i–j). Two hydrophobic residues, M377 of the D1 ISS and V647 of the D2 ISS, are inserted into a hydrophobic pocket of the adjacent ATPase domains, respectively (Fig. 5i–j). In contrast, in relatively loose interfaces, such as the P1-P2 interface in the D1 ring and the P6-P1 interface in the D2 ring, the ISS adopts a helical form for its N-terminal part, and M377/V647 is released from the hydrophobic pocket of the adjacent ATPase domains (Fig. 5i–j). The loop or helical conformation correlates perfectly with the compactness of the active

center, and only in the loop conformation, the enzymatic center is in an active state, with proper configuration for the AFs (D1:R401/R404; D2:R671/R674) and a few ATPase center-surrounding residues, including D375/D645 from the ISS, R297/K568 from the downstream sequence of Walker A motif (Fig. 5i–j). These structural observations clearly suggest the role of ISS in regulating ATPase activity in a conformation-dependent manner.

To verify the functional relevance of the two residues (M337 and V647) in the ISS motifs, we tested the effects of M337R and V647R mutations on cells using spotting assay and ribosome profile analysis. V647R severely inhibited the cell growth in an extent similar to that of the Y590A pore-loop mutant, while M337R displayed no detectable difference. Ribosome profile analysis further confirmed the importance of V647R in the in vivo function of Drg1 (Fig. 6c, d). Importantly, these results are consistent with the pore-loop mutational data (Fig. 6a, b) that the D2 domain is more important for the function of Drg1.

Altogether, our data suggest that Drg1 possesses a highly conserved ISS in both the D1 and D2 domains, and the two ISS undergo a helix-loop conformational switch during peptide processing.

## Benzo-diazaborine binds to the active centers of both the D1 and D2 domains

The cryo-EM map of Drg1 hexamers treated with ATP/benzo-diazaborine was resolved at a resolution of 3.8 Å (without symmetry imposed; Fig. 7 and Supplementary Figs. 1 and 5a–e). Notably, compared with structures from other samples, this structure is more symmetrical at both the D1 and D2 rings, and the six protomers are roughly in a same conformation (Fig. 7d). Importantly, analysis of the benzo-diazaborine occupancy in the hexamers of the planar and helical states from the ADP/AMPPNP/benzo-diazaborine dataset led to a similar observation. Benzo-diazaborine was detected in the six D1 ATPase sites of the planar state (Supplementary Fig. 9), which is more symmetrical, whereas all the ATPase sites in the helical hexamer show no trace of benzo-diazaborine. Therefore, it is apparent that the treatment of Drg1 hexamers with benzo-diazaborine led to the formation of more symmetrical structures, indicating that the chemical also directly modulates the conformation of Drg1 hexamer, in addition to the inhibition of the ATP hydrolysis.

Benzo-diazaborine is docked in equivalent sites of the D1 and D2 domain (Fig. 7f, g). On the D1 domain, benzo-diazaborine is surround by I422, T458, R297, and R429 (Fig. 7f). Importantly, benzo-diazaborine contains an extra benzyl group, which is coordinated by R429 through a cation-π interaction. Since thieno-diazaborine does not have this aromatic ring, this may explain why only the D2 domains are occupied by thieno-diazaborine in the previous Drg1 structure[44]. The binding position of benzo-diazaborine on the D2 domain similar as the one reported for thieno-diazaborine[44], and agrees with previous genetic data that yeast strains harboring Drg1 mutations of A569V, C561T, and V725L/E are resistant to diazaborine treatment[33]. In the D2 domain, the equivalent position of R429 is K695, which might also contribute to the stabilization of the benzyl group. The boron atom of diazaborine forms a covalent bond with the 2′-OH of the ribose of NAD$^+$ in the structure of FabI-diazaborine complex[56,57]. Therefore, benzo-diazaborine should have also formed covalent bonds with the 2′-OH of the ribose of the ATP molecules in the ATPase centers of both the D1 and D2 domains (Fig. 7f, g).

Next, we performed pairwise comparison of the subunit of the Drg1-ATP/benzo-diazaborine (Drg1-ATP/benzo-diazaborine) complex with the six protomers of the double mutant Drg1-ATP complex. While the NTD and D1 remain largely as a rigid body in all these comparisons, the D2 domain of the Drg1-ATP/benzo-diazaborine complex displays a distinct conformation that is dissimilar to any of the six protomers from the mutant Drg1-ATP complex (Supplementary Fig. 14). These results suggest that benzo-diazaborine has locked the ATPase centers of Drg1 in a specific conformation and limited its conformational

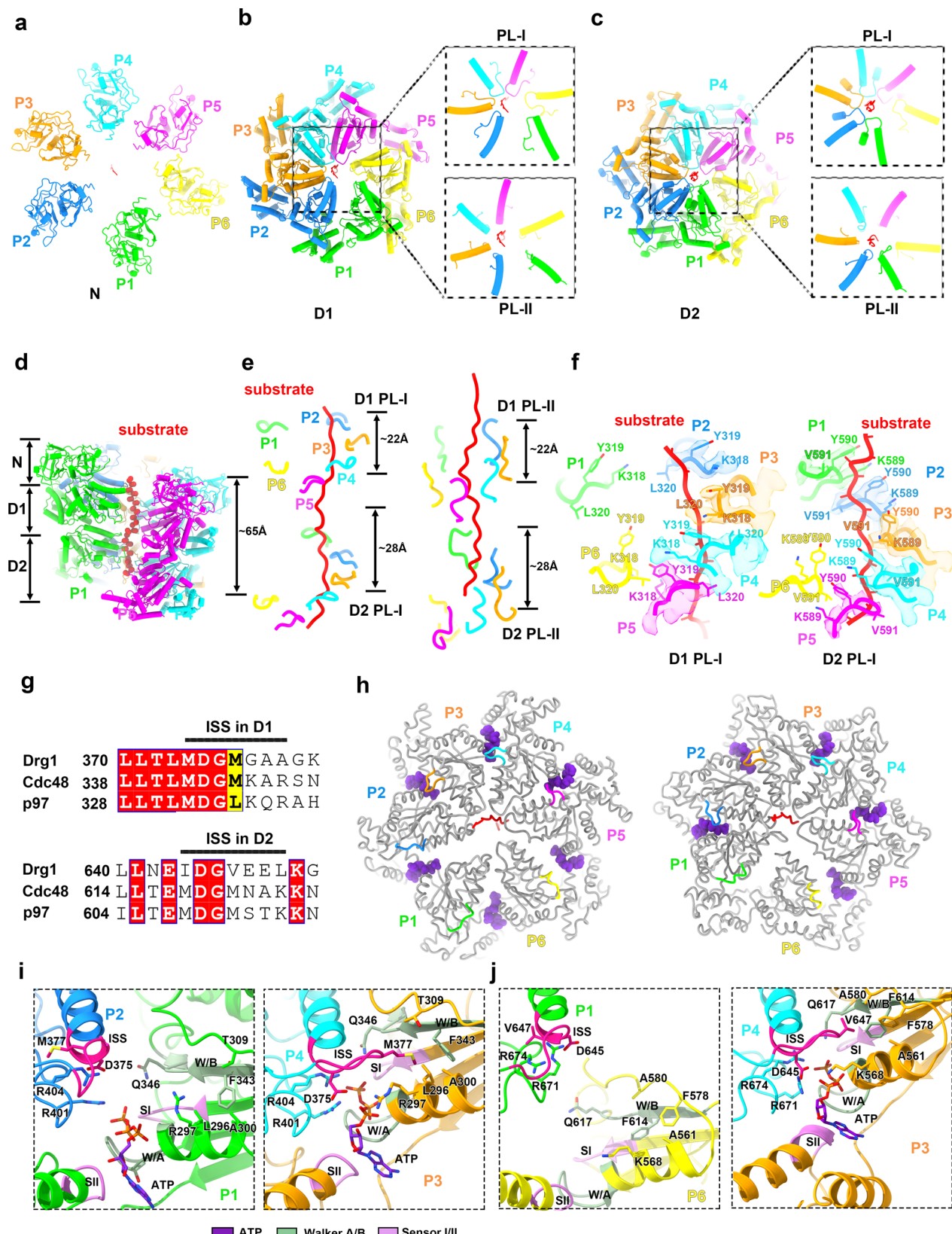

transition. Mechanistically, benzo-diazaborine probably has impaired both the intra-ring and inter-ring communications of Drg1 hexamers.

In summary, benzo-diazaborine blocks the ATP hydrolysis cycle and hinders the conformational changes of the D1 and D2 rings, thereby disabling the conformational cycling of Drg1 subunits required for substrate processing.

**Functional relevance of the NTD-D1 and D1-D2 linkers of Drg1**

Structural comparison between the helical and planar conformations of Drg1 shows a large rotation between D1 and D2 rings (Supplementary Figs. 6 and 14). Also, we noticed that, in our structures, the NTD lacks large-scale conformational difference, very different from that of Cdc48/p97[8,11,19,58,59]. These observations imply a possible role of the

**Fig. 5 | Structure of the Drg1 hexamer in processing a peptide substrate. a** The atomic model of the NTD ring of the Drg1 hexamer in the substrate-processing state. The six protomers are indicated as P1 to P6. **b** The atomic model of the D1 ring shown in cartoon representation. The PLs (PL-I and PL-II) engaged with the substrate are shown and highlighted in zoom-in panels. **c** The atomic model of the D2 ring. The PLs (PL-I and PL-II) engaged with the substrate are shown and highlighted in zoom-in panels. **d** Side view of the Drg1 hexamer with a substrate engaged in the central channel. For clarification of the substrate, P1 and P6 are not shown. **e** The spiral configuration of the PLs of Drg1 protomers around the substrate in the central channel. The interaction patterns of the PL-Is and PL-IIs are shown in the left and right panels, respectively. The vertical distance of PLs between the lowest and highest protomers are shown as indicated. **f** The atomic interactions between the PL-Is and the substrate. The conserved PL-Is (K318-Y319-L320) of the D1 domains

(left) from P2 to P5 are shown in the left panel, and the PL-Is (K589-Y590-V591) of the D2 domains from P1 to P5 are shown in the right panel. Both the atomic models and cryo-EM of the PL-Is are shown, and color-coded as indicated. **g** Sequence alignment of the ISS motifs from *S. cerevisiae* Cdc48, Drg1 and *H. sapiens* p97. **h** The ISS motifs in the D1 (left) and D2 (right) rings. The ISS motifs are color-coded, and the nucleotides are highlighted in purple sphere models. The ISS motifs of P3, P4, and P5 in the D1 ring, and P2, P3, P4, and P5 in the D2 ring form a stretched loop inserting into the neighboring nucleotide-binding site. **i** Conformations of the ISS motifs in representative tight/loose protomer interfaces of the D1 ring. The ISS motif of P2 in the D1 ring is in a retracted (helical) conformation, whereas the P4 ISS motif in the D1 ring is in a stretched loop conformation. **j** Same as **i** for the ISS motifs in the D2 ring.

---

domain linkers in modulating the structural dynamics and function of Drg1. Therefore, we analyzed the conformation differences of the linkers between the NTD and D1 and between the D1 and D2 in these structures.

The NTD-D1 linker of Drg1 is very different from that of Cdc48/p97 and contains seven more residues (Supplementary Fig. 11c). In our structures, the NTD and D1 form a relative rigid interface. In fact, the NTD-D1 linkers adopt generally similar conformations in all Drg1 protomers from the substrate-engaged to benzo-diazaborine bound hexamers. This linker is well resolved in our maps (Fig. 8), and based on the structural alignment (with the D1 as reference), the loop is in a completely different conformation from that of p97 (Fig. 8a, b). As a result, the NTD of Drg1 display a sharply different orientation from that of Cdc48/p97 (Figs. 4 and 8a, b). A few linker residues of Drg1 display highly specific contact with the NTD or the D1 to stabilize this interface. Y236 of the linker establishes hydrophobic interactions with the NTD, whereas E240 interacts with R429 of the D1 via polar interaction. P241 of the linker is seen to interact with N107 (NTD) and N301 (D1) through its main-chain atoms. These molecular interactions should contribute to the fixed orientation of the NTD on the D1 ring. Therefore, we introduced point mutations to these three positions (Y236R, E240A, P241A). Y236R and E240A were designed to disrupt the hydrophobic and polar interactions, respectively. P241A was to introduce a change to the geometry of the backbone. However, none of these mutants affected the cell growth under overexpression condition. This result suggests that these linker residues are not critical for the function of Drg1, although they may have a role in stabilizing the NTD on the D1 irrelevant to the nucleotide binding states of the D1. Notably, this is very different from that of p97, the NTD-D1 linker was reported to undergo a conformational switch to regulate the ATP binding to the D1[60].

Unlike the NTD-D1 linker, the D1-D2 linker of Drg1 is highly conserved (Supplementary Fig. 11c). Again similar to that of p97[14], this linker displays different conformations in the substrate-engaged and benzo-diazaborine bound hexamers. In the structure of the polypeptide engaged Drg1 hexamer, the linkers of P1, P2, P3, P4, P5, and P6 are in generally similar conformations, Although the linker of P1 is slightly different to a certain extent due to a relatively rotation between the D1 and D2 in P1 (Supplementary Fig. 12d). P6 is the flexible "seam" subunit and its linker was not fully resolved, but the N-terminal part adopts a helical conformation. Comparison of them with the D1-D2 linker from benzo-diazaborine treated hexamer (Fig. 8d–i) could identify two extreme conformations for the D1-D2 linker. Taking the P2 of the polypeptide-engaged hexamer as an example, the linker assumes a helical conformation. R499 interacts with the carbonyl oxygen of I506, and R504 of P2 interact with selected residues from P3, including R400, D406, and E408. I506, M503 (L464 in p97) and L508 are packed against the D2 through extensive hydrophobic interactions (Fig. 8e, f). In sharp contrast, in the protomer of planar benzo-diazaborine bound hexamer, the linker is in a loop conformation (Fig. 8g–i). Especially, M503 in the loop conformation is released from the hydrophobic interface.

This helix-to-loop conformational switch was also observed for the D1-D2 linker of p97[14,61], and the disruption of the D1-D2 linker in p97 was shown to impair its function[14,61–63]. Specifically, an over 15-Å shift of L464 (equivalent to M503 of Drg1) was observed between the structures of non-translocating and translocating p97, and L464A mutation of p97 greatly decreased the unfolding activity of p97[14]. Based on the structural analysis, we introduced four separate mutations (R499A, M503A, R504A, and F507A) to the D1-D2 linker of Drg1. The spotting assay results showed that M503A has the most apparent effect in inhibiting the cell growth, followed by R504A (Fig. 6c). The other two mutations had no effect. Further polysome profile analysis confirmed that the overexpression of M503A and R504A both lead to a "halfmer" phenotype (Fig. 6d).

Thus, our structural and functional data reveal another shared regulatory element between Drg1 and p97/Cdc48, suggesting that Drg1 might be an unfoldase in nature.

## Discussion

Drg1 is a type II AAA + ATPase that plays an essential role in the maturation of the cytoplasmic pre-60S particles. Through the purification of a double mutant form of Drg1, we accidentally captured a form of Drg1 hexamer in processing a polypeptide. The structure shows that Drg1 hexamer displays a spiral conformation, with six protomers arranged in a staircase-like configuration. Five of the six protomers directly interact with the polypeptide, with the seam subunit P6 relatively detaches from the polypeptide (Figs. 2 and 5). Substrate interactions are mediated by the four sets of PLs on the D1 and five sets of PLs on the D2 domains (Fig. 5). The structural similarity between Drg1 and Cdc48/p97 hexamers indicates that Drg1 protomers employ a conserved mechanism to grip and pull the substrate one after another during the ATPase cycle. In general, when the ATP in the P5 nucleotide binding pocket is hydrolyzed to ADP, its conformation will be converted to the P6 state and detaches from the substrate, ready to bind to ATP. When P6 binds to ATP, it will be converted to the P1 state and binds with an axial rise to the substrate.

These observations suggest that Drg1 is likely a protein unfoldase in nature, rather than a disassemblase such as NSF, which relies on large-scale NTD movement to separate SNARE complex[20]. It is possible that the removal of Rlp24 from the pre-60S particles by Drg1 also involves a threading of Rlp24. This hypothesis is supported by many experimental observations. The NTD of Drg1 closely resembles that of Cdc48/p97 despite their low sequence homology (Fig. 4). Sequence alignment between Drg1 and Cdc48/p97 also identified highly conserved functional and regulatory motifs in the D1 and D2 (Supplementary Fig. 11). We showed that the PL-I and PL-II of the D2 are more important for the in vivo function of Drg1, whereas the PLs of the D1 only play supporting roles. Similar to effects of the PL mutations, we found that only the integrity of the ISS motif on the D2 is essential for Drg1 function, and the D2 ISS acts as a switch between a helical and a loop conformation to regulate the ATPase activity of the D2. All these functional and structural features were shared by p97/Cdc48 and

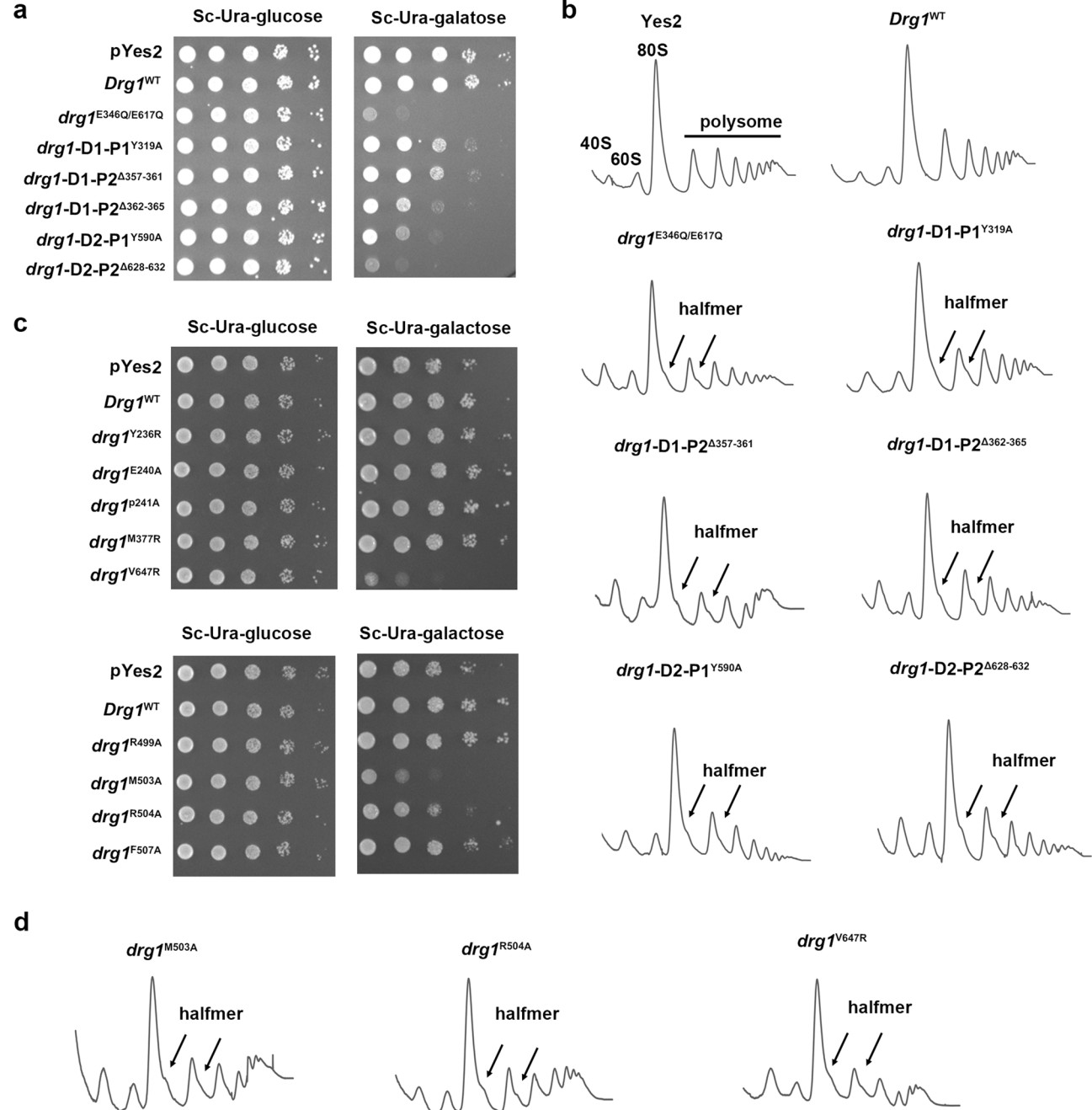

**Fig. 6 | Critical residues of the PLs, the D1-D2 linker and the ISS motifs are important for the in vivo function of Drg1. a** Drg1 pore-loop mutants suppress cell growth. Yeast BY4741 cells carrying different Drg1 variants (E346A/E617A, Y319A, Δ357-361, Δ362-365, Y590A, Δ628-632) were grown to mid-log phase in liquid medium, diluted, and spotted on SC-Ura medium with 2% glucose or galactose and incubated at 30 °C. **b** Drg1 loop mutants impairs ribosome assembly and lead to a "halfmer" phenotype. Yeast BY4741 cells expressing different variants were lyzed and the lysates were subjected to a sucrose gradient-based polysome profiles analysis. The position of 40S, 60S, 80S, and polysome peaks are indicated.

Halfmer peaks on the right shoulders of the 80S and polysome peaks are indicated by arrows. **c** The mutations on the ISS motif of the D2, and on the D1-D2 linker lead to a slow-growth phenotype. Yeast BY4741 cells carrying different Drg1 variants (R499A M503A, R504A, F507A, E240A, Y236A, P241A, V647R, and M377R) were grown to mid-log phase in liquid medium, diluted, and spotted on SC-Ura medium with 2% glucose or galactose and incubated at 30 °C. **d** Sucrose gradient polysome profile analysis of yeast cells expressing different Drg1 variants (M503A, R504A, V647R).

other AAA + unfoldases[2,11,12,14,23]. Furthermore, Drg1 also contains a conserved D1-D2 linker, which again exhibits two translocating and non-translocating conformations (helix vs. loop), similar as p97[14,61]. Most recently, Prattes et al. assembled a complex of pre-60S particle and Drg1 in vitro and reported a structure for the extraction of Rlp24 from pre-60S particle by Drg1[64]. They found that the C-terminal tail of Rlp24 is threaded into the central pore of Drg1, and translocated by the PLs of Drg1 via a hand-over-hand translocation mechanism. These

findings are consistent with our structures of the polypeptide-engaged Drg1 hexamers.

The NTD of Cdc48 binds to diverse adapter proteins to recognize different substrates in multiple physiological processes[65]. Drg1 may work similarly as Cdc48 does and requires an adapter protein. Previously, it was reported that several nucleoporins of nuclear pore complex, including Nup116, could interact with Drg1, and Nup116 could facilitate the release of Rlp24 from pre-60S particles in vitro[32].

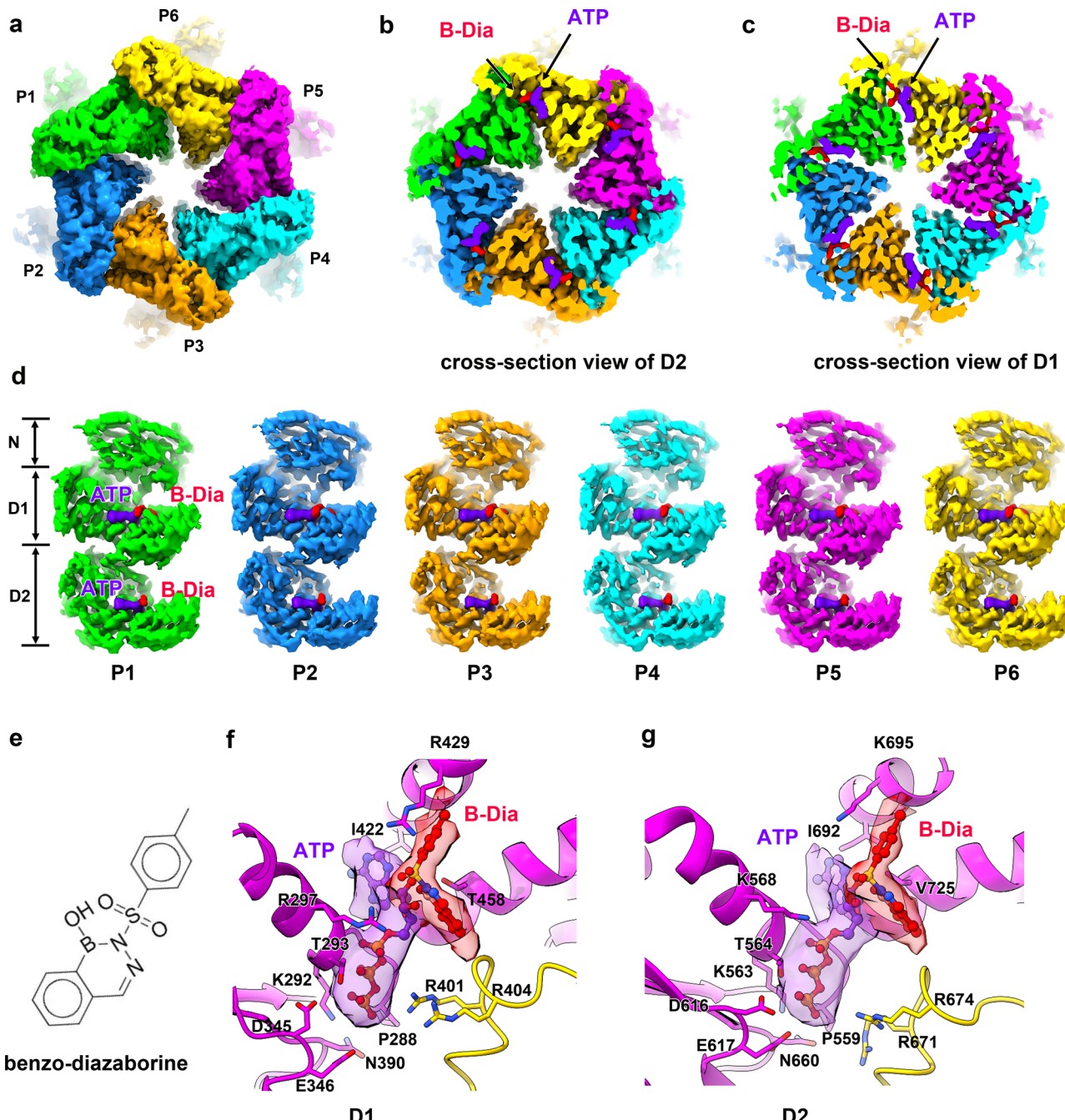

**Fig. 7 | Benzo-diazaborine binds to the ATPase sites of both the D1 and D2 domains. a** The cryo-EM map (without imposing symmetry) of the Drg1 hexamer in the presence of ATP and benzo-diazaborine (B-Dia) viewed from the bottom (D2 direction). Subunits of the Drg1 hexamer are colored separately. **b**, **c** Cross-section view of the D2 (**b**) and D1 (**c**) rings. B-Dia and ATP are colored red and purple, respectively. **d** Comparison of the six protomers, indicating that these protomers adopt a similar conformation. The binding sites of B-Dia and ATP in the nucleotide binding pockets are shown as indicated. **e** The chemical structure of B-Dia. **f**, **g** Zoom-in views of the ATPase sites of the D1 (**f**) and D2 (**g**) domains. Functionally important residues of Walker A (D1: P288, K292, T293; D2: P559, K563, T564), Walker B (D1: D345, E346; D2: D616, E617), Sensor I (D1:N390, D2: N660) and arginine fingers (D1:R401, R404; D2: R671, R674), and residues involved in interacting with B-Dia (D1: I422, T458, R429, K297; D2: I692, V725, K695, K568) are indicated.

Most recently, it was found that Arx1 and ES27 on the pre-60S particle forms a docking platform to interact with the NTD of Drg1 to recruited it to the pre-60S particle[64]. Whether any of them is a bona fide adapter merits further investigation.

Diazaborine was first discovered as an antibacterial compound that blocks the biosynthesis of fatty acids and phospholipids, thereby preventing bacterial proliferation[40,41]. Diazaborine inhibits the activity of enoyl reductase by forming a covalent bond between the 2′ hydroxyl of nicotinamide ribose and boron atom (Baldock et al.). Later, it was

found that *S. cerevisiae* was also sensitive to diazaborine and a few potential target genes including *PDR1*, *PDR3*, and *DRG1* were identified[39]. More recently, a diazaborine derivative was demonstrated to be a potent and selective inhibitor against human neutrophil elastase, which is a promising therapeutic target for many inflammatory diseases[66]. This broad-spectrum activity of diazaborine compounds highlights its high therapeutic potential. Over years, there has been an continuous effort of synthesizing and evaluating new classes of diazaborines for different purposes[67]. There are several types of

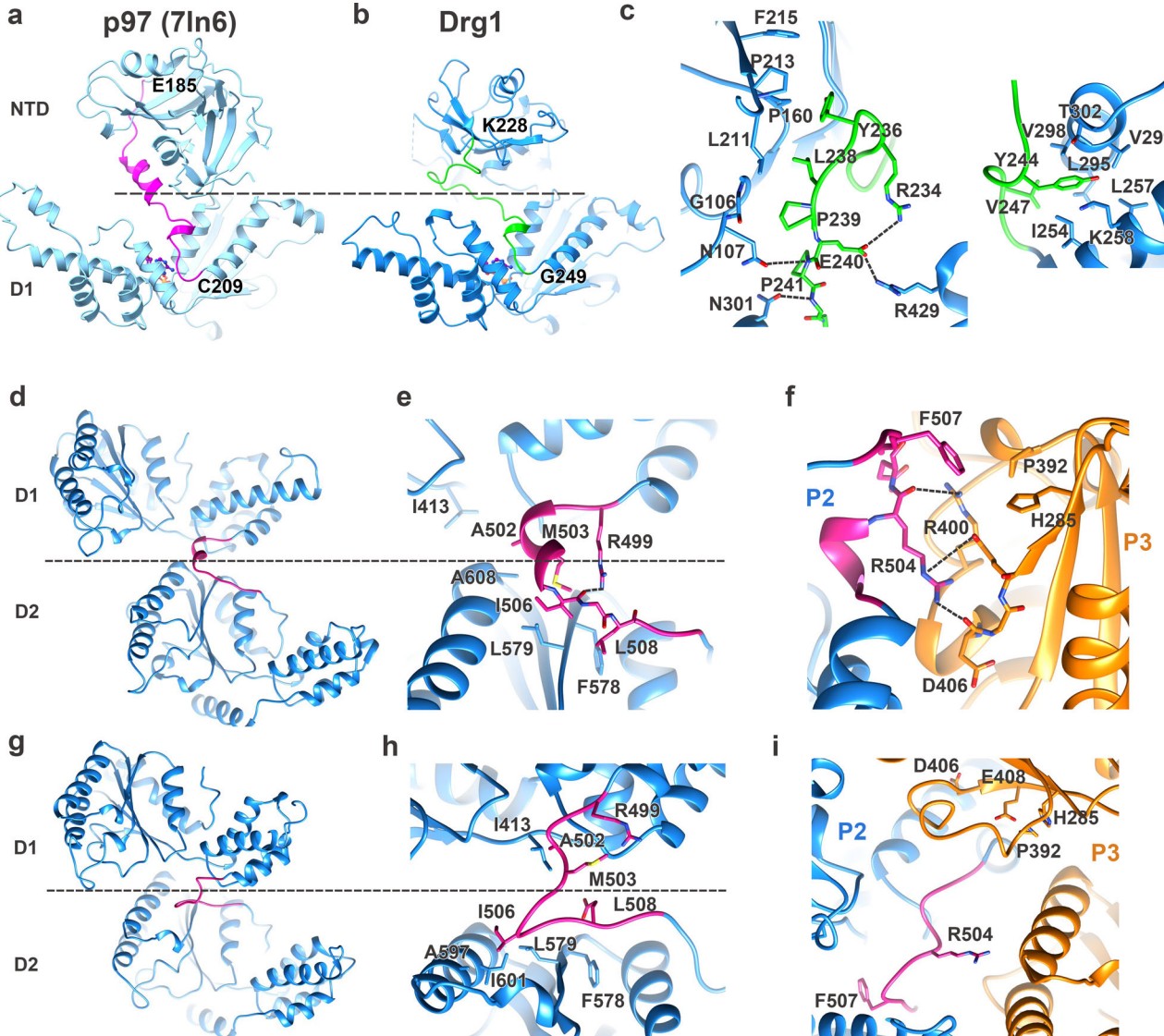

**Fig. 8 | Structural comparison of the NTD-D1 and D1-D2 linker in the Drg1 hexamers. a, b** Structural comparison of the NTD-D1 linker in the structures of p97 (**a**) and Drg1 (**b**). **c** Interaction details of the NTD-D1 linker in the structure of Drg1 hexamer. **d**–**f** Structure of the D1-D2 linker in the substrate-engaged (translocating) Drg1 hexamer. Detailed interactions are shown in **e**, **f**. The D1-D2 linker is shown in purple. **g**–**i** Structure of the D1-D2 linker in the benzo-diazaborine bounded (non-translocating) hexamers. Detailed interactions are shown in **h**–**i**.

diazaborines, including benzene, naphthalene, thiophene, furan, and pyrrole. In this work, we discovered that benzo-diazaborine binds to both the D1 and D2 domains of Drg1. Benzo-diazaborine contains a benzyl ring instead of propyl group in thieno-diazaborine, and this aromatic group indeed form a strong cation-π interaction with R429 of the D1 domain (Fig. 7). This readily explains why benzo-diazaborine could also bind to the D1 domain, whereas thieno-diazaborine prefers the D2 domain[44].

More importantly, our data show that the binding of benzo-diazaborine to the D1 and D2 sites inhibits the nucleotide-driven conformational changes on the Drg1 hexamer, locking the individual subunits in a conformation different from those typical ones seen in the substrate-processing hexamer (Supplementary Fig. 14). The conformational modulation benzo-diazaborine on the Drg1 hexamer is that it allosterically impairs the inter-subunit and inter-ring communications, resulting in a more symmetrical configuration for the hexamer. Similarly, UPCDC30245 is a conformation-selective inhibitor that preferentially binds the ADP state of p97 at the interface of the D1 and D2 domains, and the mechanism of UPCDC30245 action was reported to inhibit D1-D2 conformational communication[8]. The same observation was also found in the structure of p97 bound with a different inhibitor CB-5083[59]. Being an ATP analog competing for the ATP binding to the D2 domain of p97, CB-5083 also blocks inter-subunit and inter-domain communications to exert its function[59]. Our data show that benzo-diazaborine could bind to both D1 and D2 domains, suggesting that it might be a more potent anti-fungal agent than other diazaborine classes. Thus, further development could be focused on the modification benzyl group to increase its affinity for Drg1 to treat fungi infection. Additionally, our structural data and previous studies on the structures of diazaborine derivatives bound with different target proteins[44,56,66,68] together lay a foundation for tailoring diazaborine compounds to target various enzymes including eukaryotic AAA + ATPases for therapeutic or other purposes.

## Methods
### Protein expression and purification
*S. cerevisiae* genome was isolated from yeast strain BY4741 using LiOAc-SDS method. Full-length *AFG2* that express Drg1 was cloned into

pET28a vector with a N-terminal His6-SUMO tag followed by a TEV protease cleavage sequence. The expression plasmid of wild type Drg1 was used as template to prepare the double mutant E346Q/E617Q by site-specific mutagenesis. The proteins were overexpressed in *E. coli* Transetta (DE3) cells (TRANS). Protein purification methods were adapted from a previous work[19]. Briefly, cells were grown at 37 °C to an $OD_{600}$ of 0.6–0.8 in LB medium with 50 μg/mL Kanamycin and 25 μg/mL chloramphenicol and induced with 0.5 mM IPTG at 18 °C overnight. Cells were harvested and resuspended in lysis buffer (50 mM Tris-HCl, pH 8.0, 300 mM KCl, 5 mM $MgCl_2$, and 40 mM imidazole). The suspension was supplemented with 1 mM PMSF and 1% TritonX-100, and cells were lysed by sonication. Lysates were cleared by centrifugation in a JA-25.50 rotor (Beckman) for 75 min at 30,970×$g$ and the supernatants were incubated with Ni−NTA agarose beads (GE Healthcare) for 2 h at 4 °C. Beads were transferred into gravity column and washed with lysis buffer. Subsequently, the resins were washed with lysis buffer containing 80 mM imidazole, and proteins were eluted in elution buffer (50 mM Tris-HCl, pH 8.0, 300 mM KCl, 5 mM $MgCl_2$, and 500 mM imidazole). The eluates were supplemented with 5 mM DTT and TEV protease, and then incubated at 18 °C for 2 h to remove His6-SUMO tags. Protein solutions were loaded onto a Mono Q column (GE Healthcare). Proteins were eluted with a linear gradient to 500 mM KCl, pooled and concentrated using 100-kDa-cutoff spin concentrator (Millipore). To prepare different samples, proteins were incubated with 5 mM nucleotide (ADP, AMPPNP, ADP/AMPPNP for Drg1[WT], and ATP for Drg1[E346Q/E617Q]) for 2 h at room temperature before size-exclusion chromatography, which was pre-equilibrated with 50 mM HEPES-KOH, pH 8.0, 300 mM KCl, 5 mM $MgCl_2$, and 5 mM DTT. Peak fractions were first examined by negative-staining electron microscopy, pooled and concentrated for cryo-EM. To obtain the benzo-diazaborine bound state of Drg1, Drg1[WT] proteins were incubated in a solution containing 5 mM ATP, 2 mM benzo-diazaborine (CAS#22959-81-5) (prepared from 200 mM stock solution in DMSO), 50 mM HEPES-KOH, pH 8.0, 300 mM KCl, 5 mM $MgCl_2$ and 5 mM DTT for 2 h at room temperature before gel filtration.

## Cryo-EM sample preparation and data collection

The sample homogeneity was first examined by negative-staining with 2% uranyl acetate. Protein samples at a concentration of 0.02 mg/mL were applied onto carbon-coated copper grids, and examined with an FEI Tecnai T20 TEM at 120 kV. Initial cryo-EM sample preparation found that Drg1 hexamers suffered from a severe dissociation. Therefore, GraFix[69] was used to counteract the sample dissociation during cryo-grid preparation. Fractions from gel filtration containing intact Drg1 hexamers were pooled and concentrated to a volume of 150 μL and applied onto a 2 mL glycerol gradient (10−30%) containing glutaraldehyde (0−0.05%) in a buffer (50 mM HEPES-KOH pH 6.8, 150 mM KCl, 5 mM $MgCl_2$, 1 mM DTT, 0.5 mM nucleotide). For the inhibitor-treated samples, 2 mM benzo-diazaborine was present in the gradient buffer and throughout the complete process. The gradient was centrifuged in a TLS55 rotor (Beckman) for 13 h at a speed of 123,600×$g$ at 4 °C. Fractions were collected manually, and the cross-linking reactions were quenched by adding 1 M Tris-HCl (pH 7.4) to a final concentration of 100 mM. Fractions containing Drg1 hexamers were examined by negative-staining electron microscopy, pooled and concentrated to remove glycerol. The final samples were exchanged into buffer A (50 mM HEPES-KOH pH 6.8, 150 mM KCl, 5 mM $MgCl_2$, 1 mM DTT) containing 1 mM nucleotide for cryo-EM analysis.

All cryo-grids were prepared with holey-carbon gold grids (Quantifoil, R1.2/1.3, 300 mesh). The grids were glow-discharged for 30 s in a plasma cleaner prior to sample freezing. Cryo-grids were prepared with an FEI Vitrobot Mark IV with the chamber set at 4 °C and 100% humidity. Grids were screened with an FEI Talos Arctica (FEI Ceta camera) before data collection. All the images were collected with FEI Titan Krios TEM (Gatan K2 summit camera) operated at 300 kV with

GIF Quantum energy filter (Gatan). The movies were acquired using the SerialEM program[70] and with a magnification of 130,000× (defocus −1.0 to −2.5 μm). The statistics of data collection was summarized in Supplementary Tables 1 and 2.

## Image processing

Motion correction and electron-dose weighting were performed using MotionCor2[71] and the contrast transfer function (CTF) parameters were estimated with the Gctf[72]. RELION (versions 3.0 and 3.1)[73] were used for particle picking, 2D and 3D classification and structural refinement. 2D templates used for particles auto-picking was generated by 2D classification of a small set of manually picked particles. The initial model was generated by RELION program from selected particles and was used as reference for the following image processing procedures.

For the sample of Drg1-ADP, 1,593,028 autopicked particles were subjected to 2D classification, resulting in a dataset of 936,387 particles for further processing. After two rounds of 3D classification, one class with more symmetrical features (class 5) was subjected to another round of 3D classification, which retained 51,928 particles for 3D refinement. The final density maps were determined at 5.9 Å (without symmetry) and 4.4 Å (with C6 symmetry imposed). To explore the structural dynamics of Drg1 hexamers, classes of good quality (classes 1, 2, 4, 5, 7, 9, and 10) were combined and subjected to one round of 3D classification. The resulting 10 classes contained comparable numbers of particles and the structures differed from each to varying extents. The dataset of Drg1-AMPPNP was similarly processed (Supplementary Fig. 3). Out of 1,823,806 auto-picked particles, 520,634 particles were selected for the final round of 3D classification and 285,167 particles were kept for final 3D refinement (final resolution 5.6 Å without symmetry).

For the sample of Drg1-ADP/AMPPNP/benzo-diazaborine (Supplementary Fig. 4), 2,205,661 particles were autopicked and 1,266,728 particles were selected for two rounds of 3D classification. One class displaying a helical conformation (class 4) was subjected to 3D refinement, resulting in a density map at 5.9 Å. Two classes of planar conformation (classes 2 and 7) were combined to generate a density map at 4.3 Å. Since the helical and planar conformations differ the most at the D2 ring, all classes of good quality (531,919 particles) were combined for another round of 3D classification into ten groups using a soft mask of the D2 ring. The ten groups were then individually refined to analyze the dynamics of Drg1 hexamers.

For the sample of Drg1-ATP/benzo-diazaborine, 1,526,542 particles were auto-picked and similarly processed. A final set of 348,781 particles with more symmetrical features was selected for the final 3D refinement (Supplementary Figs. 5a−e). After CTF refinement and particle polishing, the density map could be improved to a resolution of 3.8 Å (without symmetry). As the D1 and D2 rings showed an approximately six-fold symmetry, C6 symmetry was also tested, leading to an improved density map at 3.5 Å.

For the sample of mutant Drg1[E346Q/E617Q]-ATP, 1,064,262 auto-picked particles were processed similarly. After the final round of 3D classification, 152,973 particles with relatively stable structural features were selected for 3D refinement, generating a density map at 3.8 Å. Application of CTF refinement and particle polishing could further improve the map to a resolution of 3.6 Å (without symmetry). To get high-resolution structures for different protomers, masked 3D refinement was performed for each protomer using protomer-specific soft masks.

## Model building and refinement

The model of Drg1 was built into the map of the mutant Drg1-ATP complex using Coot[74]. The predicted 3D model from I-TASSER[75] was as the initial template. The modeling was facilitated by the secondary structure prediction using PSIPRED[76]. For other protomers in the map

of Drg1 hexamers, models of Drg1 domains were fitted as rigid bodies followed by manual adjustment and rebuilding in COOT. For the D1 domain of protomer 6 (P6) that is of low resolution, the model was only adjusted using rigid body fitting. The substrate was built using a poly-alanine model. The atomic models of hexamers from different samples were similarly obtained. The automatic model refinement (Supplementary Table 2) was performed by real-space refinement (phenix.real_space_refine) in Phenix[77]. Model validation was calculated by MolProbity[78]. Figure preparation and structure analysis were performed with and UCSF ChimeraX, Chimera[79] and PYMOL (pymol.org).

## Cloning of *AFG2* variants in yeast

Wild type *S. cerevisiae AFG2* was cloned to pYes2 vector which is under the control of GAL1 promoter. This plasmid was used as template for the construction of different Drg1 mutants (Y319A, Δ362-365, Y590A, Δ628-632, Δ357-361, R499A M503A, R504A, F507A, E240A, Y236A, P241A, V647R, and M377R). The plasmid pET28a-Drg1$^{E346Q/E617}$ was used as template for the construction of pYes2-Drg1$^{E346Q/E617}$. All these constructs were verified by DNA sequencing. Drg1 variants were transformed in *S. cerevisiae* and were overexpressed with induction of 2% galactose.

## Spotting assay

Yeast cells harboring different Drg1 variants were grown in 5 mL liquid medium over night at 30 °C. Cells for spotting assay were diluted to serial concentrations ($OD_{600} = 1.0$, $10^{-1}$, $10^{-2}$, $10^{-3}$, $10^{-4}$) with sterile water. The diluted cells were spotted on SC-Ura plates containing glucose or galactose for induction and incubated at 30 °C.

## Polysome profile analysis

Yeast strains were grown overnight at 30 °C in 20 mL SC-Ura medium containing 2% glucose. Cells were diluted to $OD_{600}$ of 0.1 and resuspended in the 200 mL SC-Ura induction medium containing 2% galactose. When the cells were grown to $OD_{600}$ of 0.8 in the induction medium, cycloheximide was added at a final concentration of 100 μg/mL immediately before harvesting, and the cultures were placed immediately in ice-water.

Cells were harvested and washed twice with 2 mL of ice-cold lysis buffer (20 mM HEPES-KOH, 100 mM KCl, 12 mM MgCl$_2$, 100 μg/mL cycloheximide and 1 mM DTT). The pellets were resuspended in 1 mL of lysis buffer. Cells were disrupted with glass beads on a Disruptor Genie (Scientific Industries) and vortex 10 times for 10 s with 20 s intervals at 4 °C. Lysate was clarified at 13,523×*g* for 20 min at 4 °C. Samples corresponding to 10 units of $OD_{260}$ nm was layered over a 12 mL 10% to 50% linear sucrose gradient (w/v). The gradients were centrifuged SW41 rotor at 260,800 xg at 4 °C for 2.5 h (Beckman ultracentrifuge). The fractions were scanned at $A_{254}$ and polysome profiles were recorded by using the Density Gradient Fractionation System.

## Statistics and reproducibility

Purification of Drg1 and preparation of hexamers were repeated for more than three times with similar results.

No statistical analysis, other than those embedded in image processing, has been applied throughout the work.

## Reporting summary

Further information on research design is available in the Nature Portfolio Reporting Summary linked to this article.

## Data availability

The cryo-EM maps (C1 symmetry) of the Drg1 hexamers from the datasets of Drg-ADP, Drg1-AMPPNP, Drg1-ADP/AMPPNP/benzo-diazaborine (helical conformation), Drg1-ADP/AMPPNP/benzo-diazaborine (planar conformation) and Drg1-ATP/benzo-diazaborine have been deposited in the EMDB, with accession codes EMD-32397, EMD-32399, EMD-32400, EMD-32402, and EMD-32403, respectively. The corresponding atomic models have been deposited in the PDB under the accession codes 7YKK, 7YKL, 7YKT, 7YKZ, and 7WD3, respectively. The cryo-EM map and atomic coordinates of the Drg1 hexamer from the Drg1 mutant sample have been deposited in the EMDB and PDB with accession codes EMDB-32396 and 7WBB, respectively.

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

## Acknowledgements

We thank the Core Facilities at the School of Life Sciences, Peking University for help with negative-staining EM; the Laboratory of Electron Microscopy and Cryo-EM Platform of Peking University for help with data collection; the High-performance Computing Platform of Peking University for help with computation; the National Centre for Protein Sciences at Peking University for assistance with protein preparation. The work was supported by the National Science Foundation of China (32230051 to N.G.), the National Key Research and Development Program of China (2019YFA0508904 to N.G.), the Qidong-SLS Innovation Fund (to N.G.), and the China Postdoctoral Science Foundation (2021T140016). C.M. was supported in part by the Post-doctoral Fellowship of Peking-Tsinghua Center for Life Sciences and by Changping Laboratory.

## Author contributions

D.W., C.M., and Q.C. prepared the samples; C.M. collected the cryo-EM data; C.M. and N.G. performed EM analysis and model building; N.G., C.M., and D.W. wrote the manuscript.

## Competing interests

The authors declare no competing interests.
