## [Peer Review File · Nature Communications]

Structural dynamics of AAA+ ATPase Drg1 and mechanism of benzo-diazaborine inhibitionREVIEWER COMMENTS

Reviewer #1 (Remarks to the Author):

In this manuscript the authors determined several cryo-EM structures of the AAA-ATPase Drg1, which plays an essential role in ribosome biogenesis. The first cryo-EM structure of Drg1 was reported last year. This paper expands upon initial structural work by determining structures of Drg1 in different nucleotide/inhibitor bound states and of a double Walker B mutant with a trapped substrate. This work reveals the structure of the Drg1 NTD, which was not ordered in the original structure. The Drg1 NTD shares structural similarity to the related AAA-ATPase Cdc48/p97, however based on the structures presented in this manuscript the Drg1 NTD appears to remain in the up conformation in contrast to Cdc48/p97, where the position of the NTD varies based upon the nucleotide bound state. This work also suggests that Drg1 is a molecular unfoldase which passes substrates through its central pore. This finding is not a surprise given the high amount of sequence and structural similarity between the D1 and D2 domains of Drg1 and Rix7/Cdc48/p97/VAT. Overall, the Drg1 structures provide new insight into Drg1's unfoldase mechanism and reveal new insight into the NTD, however my major criticism is that this paper is just cryo-EM structures and is lacking any sort of biochemical/cellular validation. I would not support publication of this manuscript at Nature Communications unless the authors can provide experiments to validate the major findings derived from the structures.

Major Comments:

1. The authors must provide validation to support their model that Drg1 is a molecular unfoldase. This could be demonstrated with in vitro unfolding assays and/or yeast-based assays that look at impacts on ribosome production with specific mutants that address some of the questions below:
 - a. Both pore-loops from Drg1 contain the classic aromatic residue found in other unfoldases. While Drg1 shares many characteristics with Cdc48 the pore loop 1 composition is different. Cdc48 does not have an aromatic motif in the D1 pore loop 1 and genetic experiments have shown that addition of an aromatic motif to this pore loop causes severe growth defects in yeast (Esaki et al Scientific Reports 2017). What is the significance of the pore-loop composition in Drg1? Is it a requirement for both pore loop 1's to have the aromatic residue for substrate gripping or is this only required in the D2 domain? It is a bit perplexing that the D1 pore loop 1 has the aromatic motif yet earlier work suggests ATP hydrolysis in the D1 domain is not essential.
 - b. Beyond pore loop 1 are the pore loop 2's that contact the structure conserved and important for Drg1 function?
 - c. Given that the NTD does not undergo large scale movements like Cdc48 is the linker b/w the D1 and NTD important for Rix7 function?
 - d. The authors highlight conformational changes in the D1-D2 linker that occur upon engagement of a substrate. Are there critical residues within this linker region?

2. Based upon my quick sequence alignment it would appear that Drg1 has an inter-subunit signaling motif (ISS) in both the D1 and D2 domains. Recent work with p97 (Pan et al NSMB, 2021) revealed that conformational changes in this motif play an important inter-subunit signaling role. The authors should investigate the ISS in their structures and see if ISS undergoes a loop to helix switch like p97.

3. The authors make comparisons with the ADP/ATP/inhibitor structures and show the nucleotide bound states but according to the methods they did not refine/deposit the coordinates for any of these structures. Only a rigid body docking was done, however several of the maps have resolutions sufficient for refinement. The authors should refine and deposit these structures in the PDB in addition to the maps.

Minor Comments.

1. The Walker B mutation does not abolish ATP hydrolysis rather it slows it down – please rephrase this statement in the main text (line 134).

2. What are the large aggregate peaks in Fig. S1? Is most of Drg1 aggregating during purification?

3. In supplemental figures S3, S4, and S5 please label the protomers on the local resolution maps.

Reviewer #2 (Remarks to the Author):

In this manuscript, Ma et al. present multiple cryo-EM structures of the AAA+ ATPase Drg1 in nucleotide-bound, substrate-bound, and inhibitor-bound states, ranging from moderate to decent resolutions. I admire the tremendous effort the authors dedicated to this work. However, the whole manuscript is overwhelming with a stack of data and does not compose the data for a mechanistic story in a logical way. One unique finding is that the authors resolved a Drg1 (E346Q/E617Q mutant) cryo-EM structure in complex with a polypeptide substrate co-purified from bacteria. This structure suggests Drg1, similar to Cdc48/p97, may function as an unfoldase, which is divergent from Drg1's canonical function involved in ribosome biogenesis.

I anticipate more biochemical and functional data to support this intriguing and novel function of Drg1, which, otherwise, may speculate it as a structural artifact from the bacterial expression system with poor physiological relevance. In the inhibitor-bound structure, bentodiazaborine binds to the ATPase sites of both D1 and D2 domains, which is in contrast to several previous studies showing diazaborine

specifically targets the D2 domain of Drg1 (Preattes, 2021, Nat Comm; Zakalskiy, 2002, JBC; Prattes, Loibl, JBC, 2014). In sum, this study determined various cryo-EM structures of the AAA+ ATPase Drg1, but those structures don't compellingly reveal a new molecular mechanism of Dgr1, thus leading to a dampened enthusiasm for this manuscript.

We sincerely thank the reviewers for their constructive comments and experiment suggestions, which are very helpful for us to improve the manuscript.

REVIEWER COMMENTS

Reviewer #1 (Remarks to the Author):

In this manuscript the authors determined several cryo-EM structures of the AAA-ATPase Drg1, which plays an essential role in ribosome biogenesis. The first cryo-EM structure of Drg1 was reported last year. This paper expands upon initial structural work by determining structures of Drg1 in different nucleotide/inhibitor bound states and of a double Walker B mutant with a trapped substrate. This work reveals the structure of the Drg1 NTD, which was not ordered in the original structure. The Drg1 NTD shares structural similarity to the related AAA-ATPase Cdc48/p97, however based on the structures presented in this manuscript the Drg1 NTD appears to remain in the up conformation in contrast to Cdc48/p97, where the position of the NTD varies based upon the nucleotide bound state. This work also suggests that Drg1 is a molecular unfoldase which passes substrates through its central pore. This finding is not a surprise given the high amount of sequence and structural similarity between the D1 and D2 domains of Drg1 and Rix7/Cdc48/p97/VAT. Overall, the Drg1 structures provide new insight into Drg1's unfoldase mechanism and reveal new insight into the NTD, however my major criticism is that this paper is just cryo-EM structures and is lacking any sort of biochemical/cellular validation. I would not support publication of this manuscript at Nature Communications unless the authors can provide experiments to validate the major findings derived from the structures.

Major Comments:

1. The authors must provide validation to support their model that Drg1 is a molecular unfoldase. This could be demonstrated with in vitro unfolding assays and/or yeast-based assays that look at impacts on ribosome production with specific mutants that address some of the questions below:

a. Both pore-loops from Drg1 contain the classic aromatic residue found in other unfoldases. While Drg1 shares many characteristics with Cdc48 the pore loop 1 composition is different. Cdc48 does not have an aromatic motif in the D1 pore loop 1 and genetic experiments have shown that addition of an aromatic motif to this pore loop causes severe growth defects in yeast (Esaki et al Scientific Reports 2017). What is the significance of the pore-loop composition in Drg1? Is it a requirement for both pore loop 1's to have the aromatic residue for substrate gripping or is this only required in the D2 domain? It is a bit perplexing that the D1 pore loop 1 has the aromatic motif yet earlier work suggests ATP hydrolysis in the D1 domain is not essential.

RE: The D1 pore-loop I (PL-I) of Drg1 (318-KYLG-321) contains a Tyr, whereas the D1 PL-Is of Cdc48/p97 (KMAG/KLAG, respectively) lack an aromatic residue. The D2 PL-I of Drg1 (589-KYVG-592) is highly similar to its D1 PL-I. In Cdc48/p97, the D2 PL-I is MWYG/MWFG, and the equivalent position is Trp instead of Tyr. As pointed out by the reviewer, Esaki et al. showed that the introduction of an aromatic residue to the D1 PL-I of Cdc48 (M288Y, M288F or M288W) conferred a lethal phenotype, whereas the Trp residue in the D2 PL-I of Cdc48 was strictly required for cell growth (W561A mutation was lethal) (Esaki et al., 2017). In our Drg1 (double-mutation) structures, the PL-Is of the D1 and D2 interact with the peptide substrate in the same fashion (Fig. 5), suggesting that they both contribute to the substrate binding and/or threading.

As suggested by the reviewer, we carried out spotting assay and polysome profile analysis to evaluate the functional relevance of the PL-I and PL-II in the D1 and D2 domains of Drg1. Since a functional unit of Drg1 is a hexamer, overexpression of defective Drg1 mutants should have dominant negative effects. Indeed, as a control experiment, overexpression of the double mutant (E346Q/E617Q) dramatically impaired the cell growth and led to a polysome defect (halfmer). The appearance of halfmers on the right shoulder of polysome peaks is a typical phenotype of ribosome assembly defect. This was also accompanied by sharply elevated 40S/60S ratio on the polysome profile, which is another sign of the 60S assembly defect.

For the D1 and D2 PL-Is, two point-mutations were designed, Y319A and Y590A. For the PL-IIs, since they do not show loop residue-specific contact with the substrate (Fig. 5 and 6), we used three deletion mutations (based on distances measured in the structure), Δ 362-365 or Δ 357-361 for the D1 PL-II, and Δ 628-632 for the D2 PL-II.

Y319A, Δ 362-365 and Δ 357-361 are all deleterious to cell growth, but their effects are not as strong as those of the D2 mutants (Y590A, Δ 628-632) (Rebuttal Fig. 1a). Polysome profile analysis confirmed that all five mutants display a generally similar “halfmer” phenotype and a high 40S/60S ratio (Rebuttal Fig. 1b), indicating that the PL-Is and PL-IIs from the D1 and D2 domains all contribute to the *in vivo* function of Drg1.

Therefore, these results suggest that the aromatic residues in the PL-Is of the D1 and D2 are both functionally important, and confirmed the previous data (Kappel et al., 2012) that the contribution from the D2 domain is larger than that of the D1 domain.

Rebuttal Fig. 1 Functional characterization of the PL-I and PL-II of Drg1, using spotting assay and polysome profile.

b. Beyond pore loop 1 are the pore loop 2's that contact the structure conserved and important for Drg1 function?

RE: The PL-IIs of Drg1 are very short and not conserved, distinct from those of p97 and Cdc48 (Rebuttal Fig.2 and Supplementary Fig.11c). In the structures, compared with the PL-Is, the PL-IIs are in a relatively larger distance away from the substrate. Therefore, we designed three short deletion mutations ($\Delta 362-365$, $\Delta 357-361$ and $\Delta 628-632$) for the PL-IIs. Both the spotting assay and polysome profile showed that the PL-IIs of the D1 and D2 are important for the *in vivo* function of Drg1 (Fig. 6 and Rebuttal Fig. 1).

Rebuttal Fig. 2 Sequence alignment of the D1 and D2 among Drg1, Cdc48 and p97.

c. Given that the NTD does not undergo large scale movements like Cdc48 is the linker b/w the D1 and NTD important for Rix7 function?

RE: The NTD-D1 linker of Drg1 is very different from that of Cdc48/p97 and contain seven more residues (Rebuttal Fig. 3, Fig. 11c). In our structures, the NTD and D1 form a relative rigid interface. In fact, the NTD-D1 linkers adopt generally similar conformations in all Drg1 protomers from the substrate-engaged and benzo-Dia bound hexamers. As a result, the NTD of Drg1 display a sharply different orientation from that of Cdc48 (Fig. 4 and Rebuttal Fig. 4).

Rebuttal Fig. 3 Sequence alignment of the NTD-linker among Drg1, Cdc48 and p97.

In our map, this linker is well resolved (Fig. 8b-c), and based on the structural alignment (with the D1 as reference), the loop is in a completely different confirmation from that of Cdc48 and p97 (Rebuttal Fig. 4).

Rebuttal Fig. 4 Structural comparison of the NTD-D1 linker in the structures of p97 (a) and Drg1 (b).

A few linker residues of Drg1 display highly specific contact with the NTD or the D1 (Rebuttal Fig. 5, Fig.8). Y236 of the linker establishes hydrophobic interactions with the NTD, whereas E240 interact with R429 of the D1 via polar interaction. P241 of the linker is seen to interact with N107 (NTD) and N301 (D1) through its main-chain atoms. These molecular interactions should contribute to the fixed orientation of the NTD on the D1 ring. Therefore, we introduced point mutations to these three positions (Y236R, E240A, P241A). Y236R and E240A were designed to disrupt the hydrophobic and polar interactions, respectively. P241A was to introduce a change to the geometry of the backbone.

Rebuttal Fig. 5 Interaction details of the NTD-D1 linker in the structure of Drg1 hexamer.

However, none of these mutants affected the cell growth under overexpression condition (Fig. 6c and Rebuttal Fig. 6). This result suggests that these linker residues are not critical for the function of Drg1, although they may have a role in stabilizing the NTD on the D1 irrelevant to the nucleotide binding states of the D1. Nonetheless, our structures are only snapshots of a dynamic functional cycle of Drg1. We can not exclude the possibility that other linker residues may contribute to the communication between the NTD and D1 when Drg1 is engaged with its native substrate.

Rebuttal Fig. 6 Functional characteristic of the NTD-D1 linker, examined by spotting assay.

In the revision, we have incorporated these data and relevant discussion has also been added (Page 14).

d. The authors highlight conformational changes in the D1-D2 linker that occur upon engagement of a substrate. Are there critical residues within this linker region?

RE: The D1-D2 linker of Drg1 is highly conserved (Rebuttal Fig. 7, Supplementary Fig.11c) to that of Cdc48/p97.

Rebuttal Fig. 7 Sequence alignment of D1-D2 linker among Drg1, Cdc48 and p97.

This linker displays different conformations in our structures (Rebuttal Fig. 8-9). In the structure of the substrate engaged Drg1 hexamer (translocating), the linkers of P1, P2, P3, P4, P5 and P6 are in generally similar conformations, although the linker of P1 is slightly different to a certain extent due to a relatively rotation between the D1 and D2 in P1 (Rebuttal Fig. 8, and Supplementary Fig. 12d). P6 is the flexible “seam” subunit and its linker was not

fully resolved, but the N-terminal part adopts a helical conformation. Comparison with the D1-D2 linker from benzo-diazaborine treated hexamer (non-translocating) (Fig. 8d-i) could identify two extreme conformations for the D1-D2 linker.

Rebuttal Fig. 8 The conformation of the D1-D2 linkers in the substrate engaged Drg1 hexamer.

Taking the P2 of the substrate-engaged hexamer as an example, the linker assumes a helical conformation. R499 interacts with the carbonyl oxygen of L506, and R504 of P2 interact with selected residues from P3, including D400, E406, and E408. I506, M503 (L464 in p97) and L508 are packed against the D2 through extensive hydrophobic interactions (Rebuttal Fig. 9, Fig. 8d-f). In sharp contrast, in the protomer of planar benzo-diazaborine treated hexamer, the linker is in a loop conformation (Rebuttal Fig. 9, Fig. 8g-h). Especially, M503 in the loop confirmation is released from the hydrophobic interface. Similar observations have been reported for the D1-D2 linker of p97. The D1-D2 linker was shown to be important for the function of p97 (Pan et al., 2021; Tang and Xia, 2016). An over 15-Å shift of L464 was observed between the structures of non-translocating and translocating p97, and L464A mutation of P97 greatly decreased the unfolding activity of p97 (Pan et al., 2021).

Rebuttal Fig. 9 The structures of D1-D2 linker in the substrate engaged (translocating) (d-f) and benzo-diazaborine bounded (non-translocating) hexamer (g-i).

Based on the structural analysis, we introduced four separate mutations (R499A, M503A, R504A and D507A) to the D1-D2 linker. The spotting assay showed that M503A has the most apparent effect in the cell growth, followed by R504A. The other two mutations had no effect. Further polysome profile analysis confirmed that the overexpression of M503A and R504A both lead to a “halfmer” phenotype (Rebuttal Fig. 10, Fig. 6c-d).

Rebuttal Fig. 10 Functional characterization of the D1-D2 linker, using spotting assay and polysome profile.

Thus, these results suggest that the D1-D2 linker of Drg1 is important to its function and a similar helix-to-loop conformation switch exists for both Drg1 and p97.

2. Based upon my quick sequence alignment it would appear that Drg1 has an inter-subunit signaling motif (ISS) in both the D1 and D2 domains. Recent work with p97 (Pan et al NSMB, 2021) revealed that conformational changes in this motif play an important inter-subunit signaling role. The authors should investigate the ISS in their structures and see if ISS undergoes a loop to helix switch like p97.

RE: Thank you. This is a great suggestion. Highly conserved inter-subunit signaling motif (ISS) are found in both the D1 and D2 domains of Drg1 (D1:374-380, D2:644-650) (Rebuttal Fig. 11, Supplementary Fig. 11c).

Rebuttal Fig. 11 Sequence alignment of ISS among Drg1, Cdc48 and p97.

Structural analysis showed that the ISS motifs of the D1 and D2 in Drg1 are involved in the inter-subunit communication and undergo highly similar conformational changes as those of p97 (Pan et al., 2021). In relatively tight inter-subunit interfaces, such as the P3-P4 interfaces in the D1 and D2 rings, the two motifs are in extended loop conformations. Two hydrophobic residues, M377 of the D1 ISS and V647 of the D2 ISS, are inserted into a hydrophobic pocket of the adjacent ATPase domains, respectively (Rebuttal Fig. 12, Fig. 5). In contrast, in relatively loose interfaces, such as the P1-P2 interface in the D1 and D2 rings, the ISS adopts a helical form for its N-terminal part, and M377/V647 is released from the hydrophobic pocket of the adjacent ATPase domains (Rebuttal Fig. 12, Fig. 5). The loop or helical conformation correlates perfectly with the compactness of the active center, indicating a possible role of ISS in regulating ATPase activity in a conformation-dependent manner.

Rebuttal Fig. 12 Conformations of the ISS motifs in representative tight/loose protomer interfaces of the D1 and D2 ring.

To verify the functional relevance of the two residues (M337 and V647) in the ISS motifs, we tested the effects of M337R and V647R mutations using spotting assay and ribosome profile analysis. While V647R plasmid severely inhibited the cell growth in an extent similar to Y590A loop mutant, M337R displayed no detectable difference. Ribosome profile analysis further confirmed the importance of V647R in the in vivo function of Drg1 (Rebuttal Fig. 13, Fig. 6c-d).

Rebuttal Fig. 13. Functional characterization of the ISS motif in the D1 and D2 of Drg1.

3. *The authors make comparisons with the ADP/ATP/inhibitor structures and show the nucleotide bound states but according to the methods they did not refine/deposit the coordinates for any of these structures. Only a rigid body docking was done, however several of the maps have resolutions sufficient for refinement. The authors should refine and deposit these structures in the PDB in addition to the maps.*

RE: Suggestion is taken. We have deposited these structures in the PDB website.

Minor Comments.

1. *The Walker B mutation does not abolish ATP hydrolysis rather it slows it down – please rephrase this statement in the main text (line 134).*

RE: The sentence has been corrected.

2. *What are the large aggregate peaks in Fig. S1? Is most of Drg1 aggregating during purification?*

RE: Yes. The large peak is the soluble aggregates of Drg1 proteins. Drg1 proteins were first purified with a Ni-NTA column, followed by a Mono Q column. Proteins were then concentrated through ultracentrifugation. Different nucleotides were next supplemented, and the hexamers were separated through a SEC column (SD200) (Supplementary Fig.1). Therefore, we are not sure whether the aggregation occurred during purification or during ultracentrifugation.

3. *In supplemental figures S3, S4, and S5 please label the protomers on the local resolution maps.*

RE: We have labeled the promoters on the local resolution maps.

Reviewer #2 (Remarks to the Author):

In this manuscript, Ma et al. present multiple cryo-EM structures of the AAA+ ATPase Drg1 in nucleotide-bound, substrate-bound, and inhibitor-bound states, ranging from moderate to decent resolutions. I admire the tremendous effort the authors dedicated to this work.

However, the whole manuscript is overwhelming with a stack of data and does not compose the data for a mechanistic story in a logical way. One unique finding is that the authors resolved a Drg1 (E346Q/E617Q mutant) cryo-EM structure in complex with a polypeptide substrate co-purified from bacteria. This structure suggests Drg1, similar to Cdc48/p97, may function as an unfoldase, which is divergent from Drg1's canonical function involved in ribosome biogenesis.

I anticipate more biochemical and functional data to support this intriguing and novel function of Drg1, which, otherwise, may speculate it as a structural artifact from the bacterial expression system with poor physiological relevance. In the inhibitor-bound structure, bentodiazaborine binds to the ATPase sites of both D1 and D2 domains, which is in contrast to several previous studies showing diazaborine specifically targets the D2 domain of Drg1 (Preattes, 2021, Nat Comm; Zakalskiy, 2002, JBC; Prattes, Lobl, JBC, 2014). In sum, this study determined various cryo-EM structures of the AAA+ ATPase Drg1, but those structures don't compellingly reveal a new molecular mechanism of Drg1, thus leading to a dampened enthusiasm for this manuscript.

RE: The inhibitor used in our work is benzo-diazaborine, containing an extra benzyl group, which is coordinated by R429 of the D1 through a cation- π interaction (Fig. 7). Since thieno-diazaborine does not have this aromatic ring, this may explain why only the D2 domains are occupied by thieno-diazaborine in the previous Drg1 structure (Prattes et al., 2021).

We thank the reviewer for his/her comments on functional experiments. In the revision, we have performed functional characterization of the PLs, ISS motifs, the NTD-D1 linker and the D1-D2 linker (Fig. 6 and Rebuttal Figures). These results show that mutation in the conserved aromatic amino acid in the PL-I (Y319A) or the deletion of PL-II of D1 (Δ 357-361) has a mild effect on cell growth. In contrast, Y590A or the deletion of the PL-II of D2 (Δ 628-632) caused severe growth defects. These results confirm the direct involvement of the pore-loops of Drg1 in its *in vivo* function.

In the revision, we also showed that the ISS motif of the D2 and the D1-D2 linker is critically important for the function of Drg1. These results, together with our structural observations, reveal a collection of conserved structural and functional features shared by Drg1 and Cdc48/p97, suggesting that Drg1 highly likely work in a similar way as Cdc48/p97 to thread/unfold a substrate polypeptide, which should be a protein component (Rlp24 or its binding partner) of pre-60S particles just exported out of the nuclear pore complex.

During the preparation of the revision, we have also tried to set up an *in vitro* system to test the function of Drg1 and to examine critical residues identified in the structures.

(1) We tried to assemble a functional complex between WT Drg1 (or mutant Drg1) and Rlp24. Previous works have reported that Rlp24 directly interacts with Drg1 via its C-terminal flexible sequence (residues 147-199) (Kappel et al., 2012; Loibl et al., 2014). For this purpose, we incubated the WT Drg1 hexamer with Rlp24⁵⁰⁻¹⁹⁹, which contains the C-terminal flexible sequence. Unfortunately, we were not able to detect a binding between these two proteins using gel-filtration (Rebuttal Fig. 14) (repeated for more than three times).

Rebuttal Fig. 14 Validation of the interaction between Drg1 and Rlp24⁵⁰⁻¹⁹⁹ using gel-filtration.

Next, a previous work reported that Drg1-E346Q (Walker-B mutation on the D1) could be efficiently pulled-down by Rlp24^{C53} (The C-terminal 53 residues) (Kappel et al., 2012). However, we still could not repeat their observation (Rebuttal Fig. 15), although several different versions of Rlp24 (GST tag, sfGFP tag) were all tested.

Rebuttal Fig. 15 Validation of the interaction between Drg1^{E346Q} and various versions of Rlp24 using gel-filtration.

(2) We suspect that an adaptor protein might be required for the binding between Drg1 and Rlp24. Previous reported potential adaptor is Nup116 (Kappel et al., 2012), a component of the yeast nuclear pore complex. Thus, we exogenously expressed and purified Nusa-Nup116¹⁻¹⁷² from *E. coli* and incubated with the hexameric Drg1. However, gel-filtration showed that there is no binding detected between these two proteins (Rebuttal Fig. 16a). We also tested whether Nup116 could form a complex with Rlp24. Unfortunately, no complex was formed upon incubation of Nusa-Nup116¹⁻¹⁷² with sfGFP-Rlp24^{C53} (Rebuttal Fig. 16b).

Rebuttal Fig. 16 Nusa-Nup116 does not form a complex with Drg1 or Rlp24.

Overall, although various versions of Rlp24 have been tested, no functional complex could be assembled *in vitro*. Therefore, we speculate that additional factors in the pre-60S particles just exported out of the NPC might be required for Drg1 to interact with Rlp24.

We have spent several months on the *in vitro* system and was unable to get any positive results. It appears to be not practical for us fulfil this goal in the time frame of a revision. We hope the reviewer agrees with us that with the functional data added in the revision, our work is a more comprehensive study on the structural and functional characterization of Drg1.

Reference

Esaki, M., Islam, M.T., Tani, N., and Ogura, T. (2017). Deviation of the typical AAA substrate-threading pore prevents fatal protein degradation in yeast Cdc48. *Sci Rep* 7, 5475.

Kappel, L., Loibl, M., Zisser, G., Klein, I., Fruhmann, G., Gruber, C., Unterweger, S., Rechberger, G., Pertschy, B., and Bergler, H. (2012). Rlp24 activates the AAA-ATPase Drg1 to initiate cytoplasmic pre-60S maturation. *J Cell Biol* 199, 771-782.

Loibl, M., Klein, I., Prattes, M., Schmidt, C., Kappel, L., Zisser, G., Gungl, A., Krieger, E., Pertschy, B., and Bergler, H. (2014). The drug diazaborine blocks ribosome biogenesis by inhibiting the AAA-ATPase Drg1. *J Biol Chem* 289, 3913-3922.

Pan, M., Yu, Y.Y., Ai, H.S., Zheng, Q.Y., Xie, Y., Liu, L., and Zhao, M.L. (2021). Mechanistic insight into substrate processing and allosteric inhibition of human p97. *Nat Struct Mol Biol* 28, 614-+.

Tang, W.K., and Xia, D. (2016). Role of the D1-D2 Linker of Human VCP/p97 in the Asymmetry and ATPase Activity of the D1-domain. *Scientific Reports* 6.

REVIEWERS' COMMENTS

Reviewer #1 (Remarks to the Author):

The authors have done an excellent job of addressing my previous concerns. The impact of the original submission was limited because it was lacking any sort of functional data on the AAA-ATPase Drg1, which is essential for maturation of the ribosome. During the revision the authors performed functional studies in yeast to confirm the significance of many key residues from Drg1 identified in their structures including the pore loops, linkers, and the ISS. The inclusion of this data supports the structural observations and nicely shows that many of these residues are critical for ribosome assembly.

Minor Suggestion:

Very recently (while this work was likely being resubmitted) another study was published revealing the structure of Drg1 bound to a pre-ribosome (Prattes et al NSMB, 2022). This work also supports that Drg1 is a molecular unfoldase. I think the authors should briefly mention this new work in their discussion.

Reviewer #2 (Remarks to the Author):

In this revised manuscript, Ma et al. provided additional functional data including the yeast spotting assay to show that some conserved structural elements in Drg1 are critical for its *in vivo* functions. Those studies complemented the pure structural work in the initial submission, and constructively improved the manuscript. The center finding of this work is that Drg1 is actually an unfoldase. Unfortunately, other than the structural insights, the current manuscript doesn't contain any direct evidence to strengthen this finding either. For example, an *in vitro* unfoldase assay showing that Drg1 unfolds protein substrates by hydrolyzing ATP would be the smoking gun. One can examine this by using a fluorescent substrate, and monitor the degradation/decay of fluorescence signals.

I noted that the authors have spent several months on the *in vitro* system trying to form Drg1 in complex with other client proteins, but without any success. I understand the potential limitations of time and resources to perform those experiments. Therefore, I endorse the publication of this manuscript. However, I suggest the authors to make some minor edits in the text.

1. the polypeptide trapped from the bacterial expression system isn't an authentic substrate.
2. I suggest the authors to use a more moderate tone when they suggest Drg1 as an unfoldase.

Lastly, I have made some edits in the abstract section as a start example.

REVIEWERS' COMMENTS

Reviewer #1 (Remarks to the Author):

The authors have done an excellent job of addressing my previous concerns. The impact of the original submission was limited because it was lacking any sort of functional data on the AAA-ATPase Drg1, which is essential for maturation of the ribosome. During the revision the authors performed functional studies in yeast to confirm the significance of many key residues from Drg1 identified in their structures including the pore loops, linkers, and the ISS. The inclusion of this data supports the structural observations and nicely shows that many of these residues are critical for ribosome assembly.

Minor Suggestion:

Very recently (while this work was likely being resubmitted) another study was published revealing the structure of Drg1 bound to a pre-ribosome (Prattes et al NSMB, 2022). This work also supports that Drg1 is a molecular unfoldase. I think the authors should briefly mention this new work in their discussion.

RE: The suggestion is well taken. Prattes et al. have found that in the structure of the pre-60S bound with Drg1 both D1 and D2 are involved in interacting with a threaded peptide (C-terminal sequence of Rlp24) within the central channel of Drg1 (Prattes et al, 2022). Their results on unfolding mechanisms of Drg1 are highly consistent with ours. In the revision we have added relevant discussion (Page19).

Reviewer #2 (Remarks to the Author):

In this revised manuscript, Ma et al. provided additional functional data including the yeast spotting assay to show that some conserved structural elements in Drg1 are critical for its in vivo functions. Those studies complemented the pure structural work in the initial submission, and constructively improved the manuscript. The center finding of this work is that Drg1 is actually an unfoldase. Unfortunately, other than the structural insights, the current manuscript doesn't contain any direct evidence to strengthen this finding either. For example, an in vitro unfoldase assay showing that Drg1 unfolds protein substrates by hydrolyzing ATP would be the smoking gun. One can examine this by using a fluorescent substrate, and monitor the degradation/decay of fluorescence signals.

RE: We thank the reviewer for this suggestion. This is indeed the goal of our future work.

I noted that the authors have spent several months on the in vitro system trying to form Drg1 in complex with other client proteins, but without any success. I understand the potential limitations of time and resources to perform those experiments. Therefore, I endorse the publication of this manuscript. However, I suggest the authors to make some minor edits in

the text.

1. the polypeptide trapped from the bacterial expression system isn't an authentic substrate.

RE: We thank the reviewer for this suggestion. We have edited the text and the “substrate” has been replaced by “polypeptide”, to reflect that fact that the polypeptide trapped inside the Drg1 channel is not the native substrate and only mimics the pattern of native substrate interactions with the pore-loops.

2. I suggest the authors to use a more moderate tone when they suggest Drg1 as an unfoldase.

RE: Suggestion is taken.

Lastly, I have made some edits in the abstract session as a start example.

RE: We thank the reviewer for these edits. We have also made similar modifications to the text in other sections of throughout the manuscript.